Resource

# World's human migration patterns in 2000–2019 unveiled by high-resolution data

Venla Niva ●[1,7] ✉, Alexander Horton ●[1], Vili Virkki ●[1], Matias Heino[1], Maria Kosonen[1], Marko Kallio ●[1,2], Pekka Kinnunen ●[1], Guy J. Abel ●[3,4,5], Raya Muttarak[6], Maija Taka ●[1], Olli Varis[1] & Matti Kummu ●[1,7] ✉

Despite being a topical issue in public debate and on the political agenda for many countries, a global-scale, high-resolution quantification of migration and its major drivers for the recent decades remained missing. We created a global dataset of annual net migration between 2000 and 2019 (~10 km grid, covering the areas of 216 countries or sovereign states), based on reported and downscaled subnational birth (2,555 administrative units) and death (2,067 administrative units) rates. We show that, globally, around 50% of the world's urban population lived in areas where migration accelerated urban population growth, while a third of the global population lived in provinces where rural areas experienced positive net migration. Finally, we show that, globally, socioeconomic factors are more strongly associated with migration patterns than climatic factors. While our method is dependent on census data, incurring notable uncertainties in regions where census data coverage or quality is low, we were able to capture migration patterns not only between but also within countries, as well as by socioeconomic and geophysical zonings. Our results highlight the importance of subnational analysis of migration—a necessity for policy design, international cooperation and shared responsibility for managing internal and international migration.

Since the 1990s, human migration has been one of the top public concerns and political agenda items in Europe and North America[1]. Millions of people have been forced to flee due to conflicts while also millions have voluntarily moved to urban areas seeking better economic prospects. Around the world, diverse environmental factors, such as droughts, floods and other natural hazards also push people to move. Most of this mobility takes place within a short distance, making internal migration the most prevalent form of migration across the globe[2]. Indeed, climate-induced migration is shown to be more common within national borders[3]. Yet, public attention tends to focus on international migration including both voluntary and forced migration.

Subnational information about the estimates of the number of migrants (and immobile persons), their origin and destination and the conditions of migration are much needed for planning of urban services and infrastructure as well as rural development[4]. Understanding migration patterns across spatial scales—including its conditions, magnitude and impact—is thus fundamental for policy design.

Whilst subnational (5 arcmin, ~10 km resolution) decadal estimates of net migration for three decades between 1970 and 2000 are available[5], global-scale data on subnational migration for more recent years are sparse. One study[6] provides a more recent estimate of migration at grid cell level for 1975–2015 (5 year interval) but the baseline data

[1]Water and Environmental Engineering Research Group, School of Engineering, Aalto University, Espoo, Finland. [2]Geoinformatics Research Group, School of Engineering, Aalto University, Espoo, Finland. [3]International Institute for Applied Systems Analysis, Laxenburg, Austria. [4]Asian Demographic Research Institute, Shanghai University, Shanghai, China. [5]Faculty of Social Sciences, The University of Hong Kong, Pokfulam, Hong Kong. [6]Department of Statistical Sciences, University of Bologna, Bologna, Italy. [7]These authors contributed equally: Venla Niva, Matti Kummu. ✉e-mail: venla.niva@aalto.fi; matti.kummu@aalto.fi

are derived from national-level birth and death data. Other estimates describe global international migration by using national-level data[7,8] or internal migration at the national level, based on national census data. These studies suffer from a long time interval between census years (typically 10 years)[9]. The coarse spatial and temporal resolutions of these data hinder the ability to conduct gridded migration trend analyses over time.

Our study aims to address these research gaps by developing a detailed annual net migration dataset, by collecting, gap-filling and harmonizing (1) comprehensive national-level birth and death rate datasets for 216 countries or sovereign states; and (2) subnational data for births (covering 163 countries, divided into 2,555 administrative units) and deaths (123 countries, 2,067 administrative units) (Extended Data Fig. 1; Methods). In doing so, we provide a detailed analysis of the spatiotemporal development of (1) the magnitude of net migration and (2) its impact on population growth over the past two decades. Firstly, for magnitude, we collected reported data from various sources to create an annual net migration dataset for 2000–2019, using national and subnational birth and death data, downscaled to 5 arcmin resolution (~10 km at the equator) with selected socioeconomic variables (Fig. 1; Methods). These data enable us to perform analyses on net migration trends and patterns from local to global scales. Our gridded net migration data allow, for instance, comparing the intensity of net migration and its trends at several administrative scales. It is also possible to analyse the types of sending and receiving areas (rural or urban) at multiple scales (regional, national (administrative 0), provincial (administrative 1) and communal (administrative 2)) over the past two decades. Indeed, there is no such systematic global-scale classification on, for instance, which urban areas are net senders and which rural areas are net receivers. Here, we present rigorous analyses of the distribution of the types of origin and destination areas.

Our paper also contributes to analysing the impact of human migration on population change. Using our annual gridded dataset to map migration in parallel with demographic and geophysical data (Extended Data Figs. 2 and 3), we were able to assess the impact of net migration on rural and urban population change at national, subnational and communal levels and across different societal and climatic conditions. Understanding the contribution of migration to population change is crucial because migration affects sending and receiving societies in various ways. In terms of economic consequences, migration influences socioeconomic development of both sending and receiving areas—for example, by increasing productivity in receiving areas and reducing income inequalities across countries through remittances[10,11]. Nevertheless, migration can also cause considerable pressure on the infrastructure and services of the receiving areas[4] and consequently exacerbate the vulnerability of migrants[12,13]. However, few empirical studies have analysed the impact of migration on population change at the global scale over the past decades. Our analysis thus provides a solid quantitative foundation towards understanding the extended societal impacts caused by migration.

## Results

We first developed a gridded global net migration dataset at annual timesteps for 2000–2019 (Fig. 1). This here-developed dataset (openly available at https://doi.org/10.5281/zenodo.7997134) was constructed from subnational (administrative 1) birth and death rate data collected across 2,555 and 2,067 administrative units, respectively (Fig. 1a–b; Methods), downscaled to 5 arcmin resolution with rasterized socioeconomic data developed in this study and finally adjusted to match the subnational data collected (Fig. 1c–f; Methods).

Our birth and death rate data revealed considerable intranational heterogeneities, particularly in large countries, such as Russia, the United States, China, Brazil and India (Fig. 1g–h). This highlights the importance of using subnational (particularly downscaled) data instead of national data (as used in ref. 6) for understanding global

population dynamics. These downscaled birth and death data then allowed us to estimate natural change in population (deaths subtracted from births) for each year and grid cell (Fig. 1i). When combined with reported annual population change over the same time period (Fig. 1j), we were able to estimate annual net migration in each grid cell (Fig. 1k) using a similar method to that of ref. 5 (Methods). Here, net migration can be either negative (more people out-migrating than in-migrating) or positive (more people in-migrating than out-migrating).

It should be noted that our data are prone to uncertainties that originate from collected subnational data but propagate to all derived data products—including birth and death rates, natural change in population, as well as net migration estimates. Subject to data availability, we performed a partial validation for our data products by comparing gridded data with subnational (mostly administrative 2 level) data (Methods; Supplementary Table 2 and Supplementary Figs. 1–6). However, this validation cannot capture areas where uncertainties may be the highest—that is, areas in which the collected census data are not available or of poor quality, for example, those suffering from sporadic census years or changing subnational administrative units. To ensure global spatial and temporal coverage, we applied a series of adjustments and corrections to the data (Methods). Nevertheless, higher uncertainties remain in some countries (such as those in Africa, the Middle East and parts of Asia) than in others (such as those in Europe and much the Americas). As proxies of original data quality, we provide the description, resolution, timespan and sources of each collected dataset in the Supplementary Data.

### Magnitude of global net migration
**Temporal dynamics of net migration depend on scale.** To assess global migration dynamics, we aggregated net migration at three spatial levels: communal (administrative level 2), provincial (administrative level 1) and national (administrative level 0) (Fig. 2). This approach allowed us to compare the magnitude of migration that occurs at different spatial levels. Our results show that migration patterns vary remarkably across nations. On a national scale over the entire 20 year study period, net migration was positive (that is, in-migration was greater than out-migration) in Australia, North America, as well as parts of Europe and the Middle East—all being areas that have attracted either asylum or job seekers or both (Fig. 2 and Extended Data Fig. 4). Net migration was negative in countries like Syria, Lithuania, Zimbabwe, Venezuela and Guyana (Fig. 2e)—in line with previous assessments from Venezuela and Syria, where millions of people have fled a humanitarian crisis and conflict[14–16] and also from Lithuania and Zimbabwe where numerous people have out-migrated in search of better economic prospects[17,18].

At a provincial level, the migration patterns reflect the prevalence of internal migration in many countries, as can be observed from both net-positive and net-negative provinces. For example, in China, the coastal areas show positive net migration while negative net migration (out-migration is greater than in-migration) was observed in many inland provinces (Fig. 2c). This is consistent with the well-founded internal migration patterns in China where labour migration is concentrated towards urban, coastal areas[19]. The same applies to many other countries, such as the United States, where urban centres are attracting people from other states and abroad[20,21]. When assessing net migration at even finer spatial detail (communal level), interesting patterns start to arise. For example, in the United States, many states with positive net migration (Fig. 2c) are characterized by mainly negative county-level net migration (Fig. 2a). See our online net migration explorer tool for more detail: https://wdrg.aalto.fi/global-net-migration-explorer/

Our dataset also allowed us to explore the temporal dynamics of net migration over the study period (2000–2019) across three administrative levels. We assessed the trend of net migration at each level over the study period (2000–2019) by using linear regression. The results follow a similar pattern as cumulative net migration where

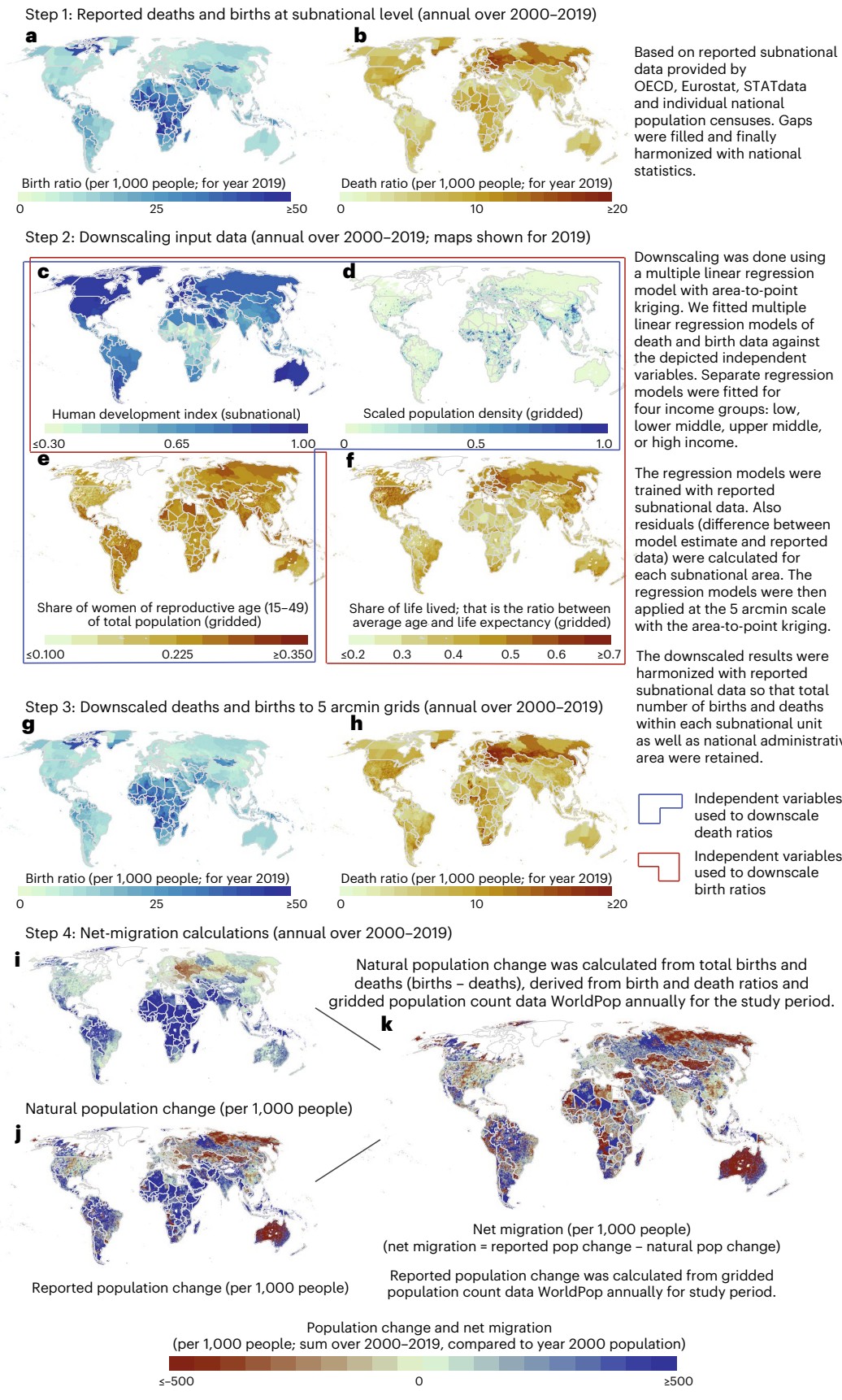

**Fig. 1 | Schematic outline of our study. a,b,** Reported annual subnational birth (**a**) and death rates (**b**) for 2000–2019 are based on various global and national datasets with some of the intermediate results (Methods). **c–f,** For downscaling to 5 arcmin grid level, we used four annual gridded or subnational datasets of (**c**) human development index, (**d**) scaled population density, (**e**) share of reproductive women and (**f**) share of life lived in each grid cell, as detailed in the panels. **g–i,** The annual downscaled birth (**g**) and death (**h**) rates allowed us to estimate the natural population change for each year (**i**). **j,k,** When combining this with reported population change based on WorldPop[55] (**j**), we were able to calculate annual net migration for 2000–2019 (**k**). See Methods for more details.

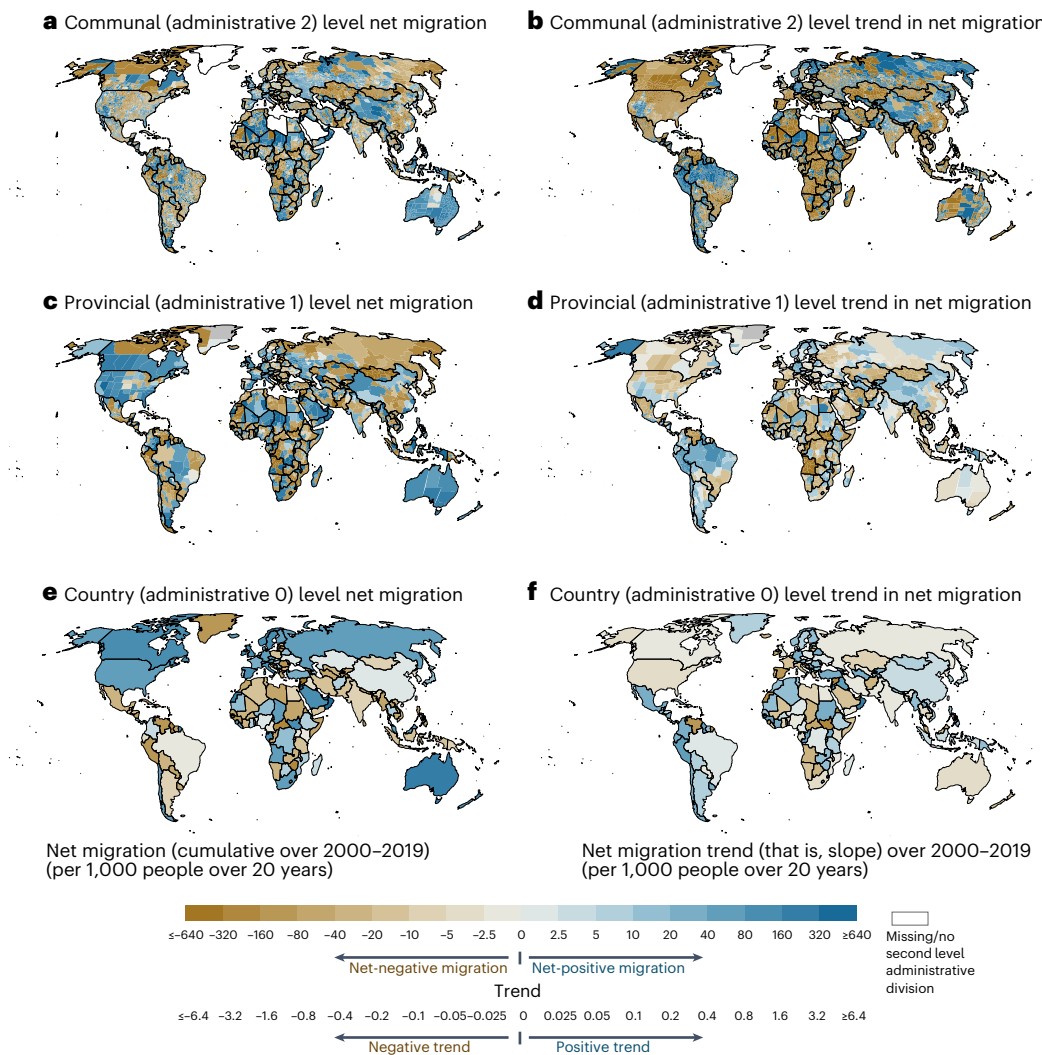

**a** Communal (administrative 2) level net migration

**b** Communal (administrative 2) level trend in net migration

**c** Provincial (administrative 1) level net migration

**d** Provincial (administrative 1) level trend in net migration

**e** Country (administrative 0) level net migration

**f** Country (administrative 0) level trend in net migration

Net migration (cumulative over 2000–2019)
(per 1,000 people over 20 years)

Net migration trend (that is, slope) over 2000–2019
(per 1,000 people over 20 years)

≤−640 −320 −160 −80 −40 −20 −10 −5 −2.5 0 2.5 5 10 20 40 80 160 320 ≥640

← Net-negative migration | Net-positive migration →

Missing/no second level administrative division

Trend

≤−6.4 −3.2 −1.6 −0.8 −0.4 −0.2 −0.1 −0.05 −0.025 0 0.025 0.05 0.1 0.2 0.4 0.8 1.6 3.2 ≥6.4

← Negative trend | Positive trend →

**Fig. 2 | Net migration for three administrative levels and cumulative net migration trends within communal, provincial and national administrative areas. a,c,e,** Sum of annual net migration over 2000–2019 is shown for communal (administrative 2) (**a**), provincial (administrative 1) (**c**) and national (administrative 0) (**e**) levels. **b,d,f,** Net migration trend (slope) over 2000–2019 is shown for communes (**b**), provinces (**d**) and countries (**f**). The trend was determined by calculating the slope of linear regression line. Negative net migration refers to a situation in which more people out-migrate than in-migrate and positive net migration refers to a situation in which more people in-migrate than out-migrate. With our online net migration explorer tool, it is possible to explore these patterns for each year and subnational unit: https://wdrg.aalto.fi/global-net-migration-explorer/.

the trend changes according to the administrative level. Further, the results show interesting patterns of where net migration has a negative trend and where it has a positive trend over the past two decades. In North America, for instance, net migration shows a declining trend in almost all regions, excluding small pockets in the southwest (Fig. 2b,d). The same applies to South America, especially Brazil and Chile, where net migration has been on a growing trend in the northern parts of these countries, while in the south the trend has been declining; and Australia, where the trend of net migration has been positive in the middle parts, while being negative in the coastal regions of the continent (Fig. 2b,d).

**Rural and urban migration show high global variation.** We further assessed the development of net migration by studying how net migration differs in rural and urban areas (Extended Data Fig. 5). Urban areas often receive migrants from rural areas—the so-called 'urban pull–rural push' situation[22]. We assessed if this holds true across 12 world regions (Extended Data Fig. 4) and for each country at three administrative levels by using the GADM delineation (national, administrative level 0; provincial, administrative level 1; and communal, administrative

level 2). Here, we combined our net migration dataset with an urban extent dataset for 2000–2019 that was created for this study. The urban extent dataset maps urban areas on the basis of scaled population density and share of urban population at national level, annually for 2000–2019 (Methods; Extended Data Fig. 3).

Our data show that when aggregated globally, urban net migration was positive (more in-migrants than out-migrants) throughout the study period (2000–2019), while rural net migration remained negative (except for years 2010 and 2012) (Fig. 3m). The magnitude of global net migration ranged annually from near-zero to around three net migrants per 1,000 people (Fig. 3m). Notable spatiotemporal variation between rural and urban net migration was evident at the regional level (Fig. 3a–l) as well as at national, provincial and communal levels (Extended Data Fig. 6). However, no considerable change in net migration rates towards urban or rural areas was observed in any region between 2000 and 2019. Both rural and urban net migration were negative (down to approximately −10 net migrants per 1,000 people) nearly throughout the study period in Central America (Fig. 3a), whereas in North America (Fig. 3g) and Oceania (Fig. 3h), total net migration was steadily positive (with a constant magnitude of at least +5 net

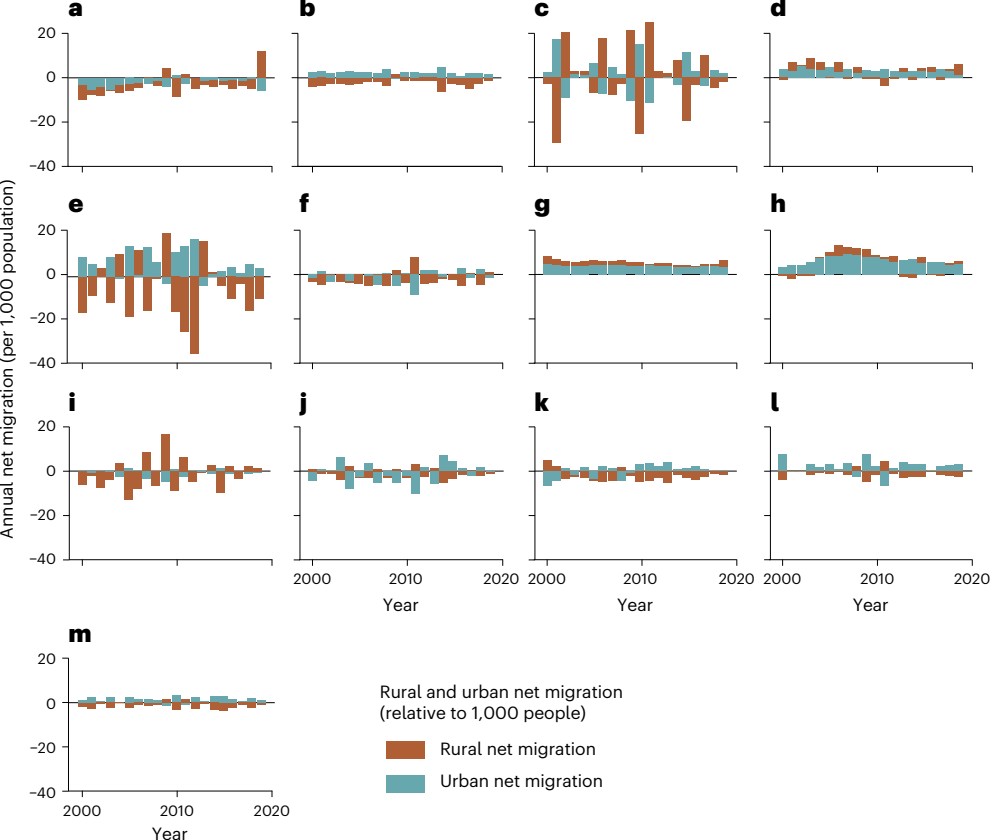

**Fig. 3 | Annual rural and urban net migration aggregated over geographical regions. a–m**, The regional sums for each year (Central America (**a**), East Asia (**b**), Eastern Europe (**c**), Europe (**d**), Middle East (**e**), North Africa (**f**), North America (**g**), Oceania (**h**), South American (**i**), South Asia (**j**), Southeast Asia (**k**) and Sub-Saharan Africa (**l**)) and the annual global sum (**m**) of urban and rural net migration. Urban and rural net migration are reported per 1,000 urban or rural inhabitants in each region. The regional division follows the UN country grouping (Extended Data Fig. 4). See gridded net migration in rural and urban areas in Extended Data Fig. 5.

migrants per 1,000 people). In East Asia (Fig. 3b), net migration was negative in rural areas and positive in urban areas (with magnitudes <5 net migrants per 1,000 people) while in other regions, the pattern was more complex (Fig. 3).

In Eastern Europe and the Middle East, rural and urban net migration rates fluctuated annually, especially during the years preceding the Arab Spring, followed by massive rural out-migration and urban in-migration between 2011 and 2013—with magnitudes up to almost +40 net migrants per 1,000 people and down to −20 net migrants per 1,000 people (Fig. 3e). Out-migration from Syria was among the largest in the world between 2010 and 2015, during when more than 2 million people left to neighbouring countries Turkey and Lebanon[2,14,23]. This explains a sharp influx of migrants to rural and urban areas in Eastern Europe group that includes Turkey (Fig. 3c). Although our results align with previous estimates of migration in the Middle East, these regions are prone to high uncertainties in the data.

**Net-receiving provinces contain a third of global population.** When focusing on national and subnational scales, the global urban pull–rural push pattern (Fig. 3m) becomes patchier (Fig. 4). At the national scale, 36% of global population (in 2019) lived in countries where this pattern was evident. These countries include the Nordics and several countries in Africa, Southeast and East Asia (Fig. 4c). However, at the subnational scale, many more people lived in provinces and communes which were either net receivers or net senders, as presented by negative or positive net migration in both urban and rural areas. Such provinces were located in the United States, Canada and Australia, while Russia, the

northeast of the United States, Mexico and the Balkans, for example, accommodated multiple net-sending provinces (Fig. 4b).

A situation where urban net migration was negative and rural net migration positive (rural pull–urban push) was observed in few locations, such as certain provinces of Indonesia, Congo, Venezuela and Pakistan (Fig. 4b), covering in total 22% of the world's population. Notably, 37% of the global population lived in the urban and rural areas of net-receiving communes. Extensive rural in-migration is probably explained by interprovincial and intercommunal migration between rural areas and immigration from other countries. Studies show that a trend of rural–urban migration is shifting towards more complex mobility patterns, of which rural–rural mobility is one of the most prevalent types of internal migration. Especially in Sub-Saharan Africa, people tend to move between rural areas as seasonal circular migration and for economic diversification, given better access to land or job prospects than in cities[24,25]. In Europe, a similar pattern appears in rural areas attracting workers in the agricultural sector within the same country or from abroad[26]. Large urban agglomerates may also push people to move to rural areas in search of more affordable housing (counter-urbanization; see ref. 27 for counter-urbanization in Australia and ref. 28 for the United States). It should be noted that the results are strongly influenced by the delineation of urban areas (Methods).

## Impact of migration on population change
**Migration often accelerates urban population growth.** Net migration taking place in densely populated areas can be relatively large compared to natural population change (Figs. 1i,k and 2a). In such

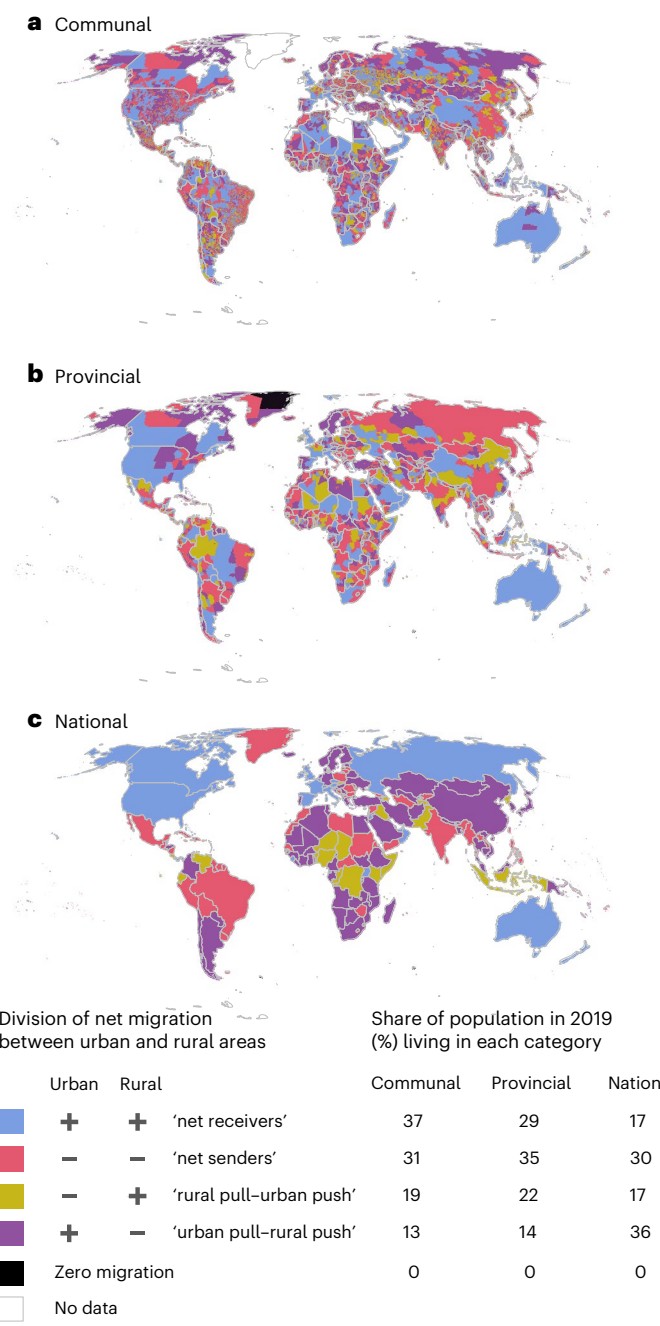

**a** Communal

**b** Provincial

**c** National

Division of net migration between urban and rural areas

Share of population in 2019 (%) living in each category

| Urban | Rural | | Communal | Provincial | National |
|---|---|---|---|---|---|
| + | + | 'net receivers' | 37 | 29 | 17 |
| − | − | 'net senders' | 31 | 35 | 30 |
| − | + | 'rural pull–urban push' | 19 | 22 | 17 |
| + | − | 'urban pull–rural push' | 13 | 14 | 36 |
| | | Zero migration | 0 | 0 | 0 |
| | | No data | | | |

+ Net-positive migration  − Net-negative migration

**Fig. 4 | Division of net migration between urban and rural areas. a–c**, The communal (**a**), provincial (**b**) and national (**c**) levels. Each administrative unit was categorized into one of the four classes on the basis of the 'direction' of migration in rural and urban areas. For example, if net migration in an administrative unit was positive in both urban and rural areas, then that unit would be categorized as a net receiver, whereas a unit in which urban net migration was positive and rural net migration negative would be categorized as urban pull–rural push. The share of population living in each category was calculated for each administrative level. For instance, 36% of global population lived in communes where both urban and rural net migration were positive (net-receiving communes). See net migration in rural and urban areas by different administrative levels in Extended Data Fig. 6.

cases, large-scale migration can strain natural and human resources, as well as infrastructure, which are insufficient to serve a steep surge in population, particularly in urban areas[4,29]. On the other hand, migration can potentially help ageing societies like those in Europe to maintain their work force[30,31]. To empirically examine the role of migration in

population change globally, we compared natural population change (deaths subtracted from births) with reported population change in rural and urban areas at the three administrative levels.

We found that about half of the global urban population lived in areas that were affected by positive net migration in a way that positive net migration added to a naturally growing urban population. Notably, in some urban areas—especially in the Nordics, Germany, Austria and Spain—positive net migration even shifted a naturally decreasing population towards growth (Fig. 5c,e). In very few countries, positive net migration slowed down natural decline in urban population but this affected urban areas accounting only for ~1% of global urban population. About 39–44% of the urban population lived in areas affected by negative net migration that slowed down urban population growth (Fig. 5). There were fewer cases (0–6% of urban population) where naturally growing urban population declined due to intensive out-migration. Such areas could be detected mainly at communal level in countries such as France, Italy, the United States and India.

Negative net migration impacted rural areas more often than urban areas: 10% of global rural population lived in communes where out-migration turned rural population growth to a total population decline (Fig. 5b). Approximately half of the global rural population lived in countries where rural population growth was slowed down by negative net migration, whereas one-fifth of rural population lived in countries where negative net migration accelerated the prevailing decline in population (Fig. 5f). Yet, positive net migration propelled rural population growth in multiple provinces and communes accounting for around a third of global rural population. In Russian provinces and communes bordering Kazakhstan, for instance, in-migration of skilled workers from Kazakhstan potentially explains the accelerated rural population growth or slowed rural population decline[32].

**Human development is associated with migration more than aridity.** Global migration is known to be driven by both socioeconomic and environmental factors[3,13]. Here, we analyse how total, urban and rural net migration co-occur with different socioclimatic conditions, dividing the globe into 100 socioclimatic bins (Extended Data Fig. 7) on the basis of the level of aridity, human development (measured by human development index (HDI); Extended Data Fig. 8) and population. Each bin accommodates ~1% of the global population (Methods).

Our results show that, over the past 20 years, high net migration often co-occurs with high HDI and high aridity, which is evident especially in urban areas (Fig. 6a). For instance, urban areas in the Arabian Peninsula, arid parts of North America, Australia, Argentina and the Mediterranean region had high (mostly above +60 net migrants per 1,000 people over the study period) positive net migration rates (Fig. 6a). High positive net-migration rates in many Middle Eastern countries, such as Qatar, Oman and Saudi Arabia (Fig. 2e), are explained by large labour migration particularly from Asian and African countries[33]. On the other hand, in Australia and North America, the 'preference of low-density living'[34] can explain positive net migration in rural areas. Notably, high HDI is common for these regions whereas the range of aridity is wider.

In terms of negative net migration, global out-migration hotspots (areas with substantial negative net migration) are located in socioclimatic conditions with middle-level aridity and human development. This was especially the case in rural areas (Fig. 6c). High negative net migration rates could be observed in regions like Central America, northeastern Brazil, Central Africa and Southeast Asia (mostly below −60 net migrants per 1,000 people over the study period). This aligns with recent studies showing that most migration is originating from areas where people have sufficient capacity to move and use migration as a form of adaptation to unfavourable environmental conditions[3].

**Positive net migration often co-occurs with high HDI.** When comparing natural population growth with net migration across different socioclimatic conditions, we found that the contribution of migration

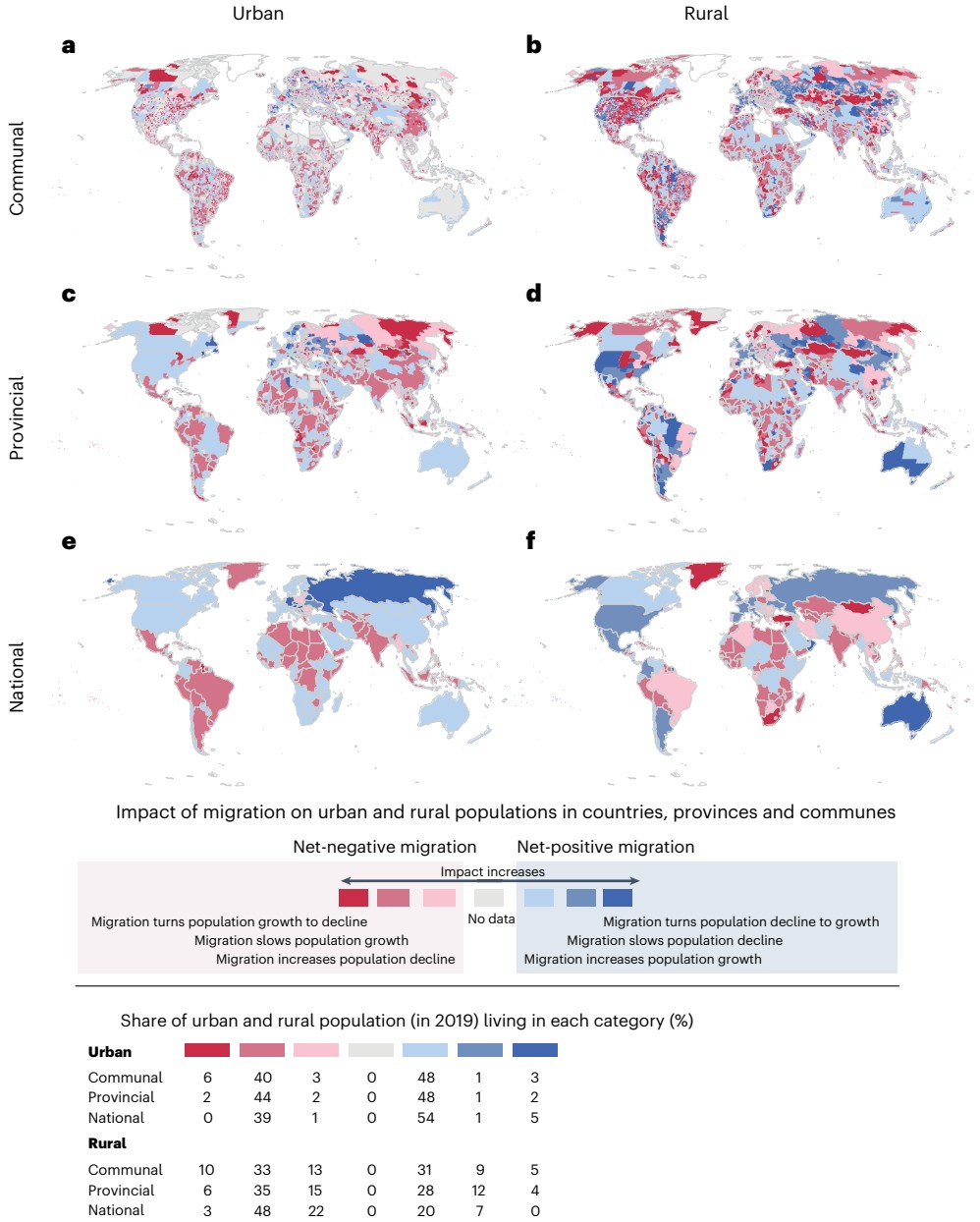

Fig. 5 | Impact of net migration on population change in urban and rural areas. a–f, The communal (a,b), provincial (c,d) and national (e,f) levels for urban (a,c,e) and rural (b,d,f) areas. Impact is divided into seven categories by comparing total population change, net migration and natural population change (growth or decline). Total population change includes both net migration and natural change. Natural change is measured with births and deaths, that is population change without migration. For example, net migration slows down urban population decline when urban net migration is positive but total urban population change is negative (Methods). Share of global urban and rural population living in each category shows the share of global urban or rural inhabitants living in areas under each type of impact.

to population change (Methods) was more strongly associated with the level of human development than with aridity (Fig. 6b,d). Positive net migration often contributed to increasing population growth, slowing population decline or shifting declining population to growth in areas with relatively high human development. Our results show that the climatic factor, as measured by the level of aridity, has a weaker association with the impact of migration on population growth. Therefore, migration can accelerate or slow down population growth in a wide range of climatic conditions.

For instance, most of Europe, North America and Australia, which are regions with high human development, experienced increasing urban population due to positive net migration (Fig. 6b). Notably, the capacity to cope with the pressure on physical and social infrastructure from population growth is high in these regions. On the other hand, many urban regions in West, East and South Africa, Arabia, as well as India, Bangladesh, China and Southeast Asia had positive net migration accelerating their natural population growth, while they also have more limited capacity (HDI) to cope with a growing population (Fig. 6b). Urban regions with the lowest human development level experienced negative net migration and consequently a slowing down of natural population growth. While this may indicate that out-migration from those areas is alleviating the pressure caused by natural population growth, it can also reflect urban out-migration pushed by urban poverty.

Finally, the role of migration in rural population change follows a general pattern in which regions with higher human development

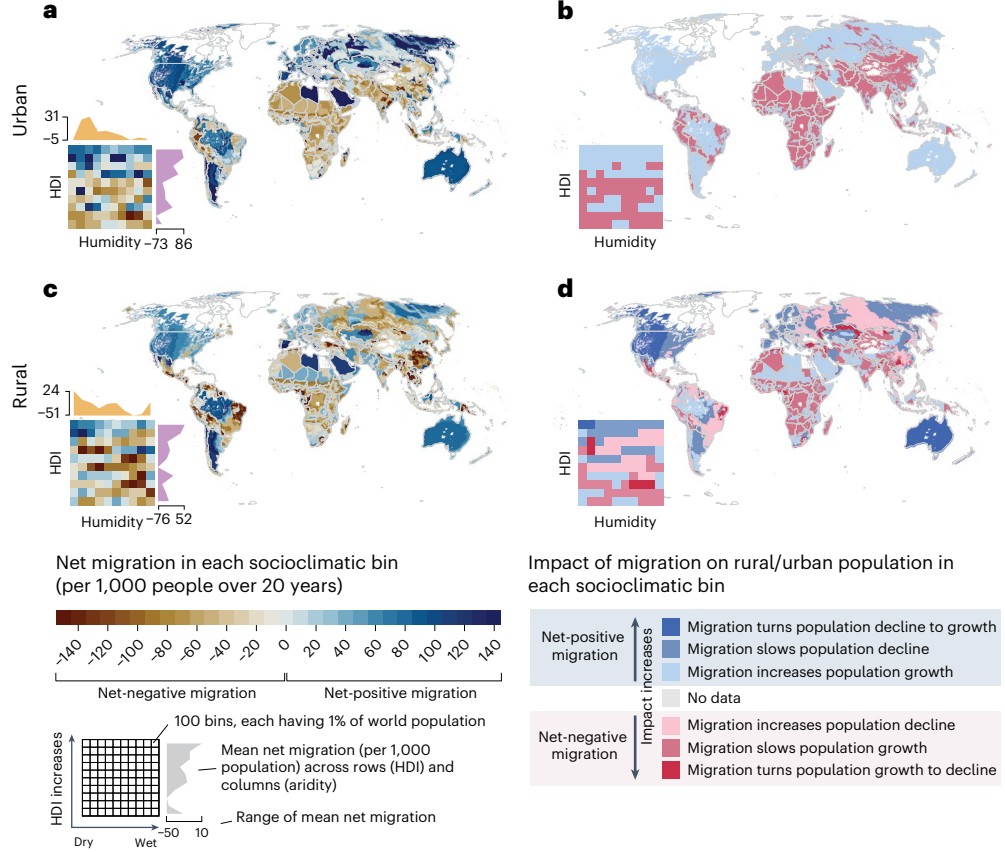

**Fig. 6 | Net migration and its impact on population change in socioclimatic bins. a–d**, For each socioclimatic bin, urban net migration per 1,000 urban population summed over the study period (**a**), the impact of urban net migration on urban population change in each socioclimatic bin (**b**), rural net migration per 1,000 rural population summed over the study period (**c**) and the impact of rural net migration on rural population change (**d**). Here, net migration per urban or rural population was calculated as a zonal sum of net migration in urban (**a**) and rural (**c**) areas in each bin and then divided with the respective urban or rural population count in the respective bin. The maps are spatial representations of the heatmaps. Socioclimatic bins are based on socioeconomic and climatic conditions represented by human development and aridity indices (Methods; Extended Data Figs. 7 and 8).

level are impacted by positive net migration. Declining rural populations turned to growth due to positive net migration in North America and Australia (Fig. 6d), whereas rural population decline was slowed down by positive net migration in most of Europe, parts of Russia and South America.

## Discussion and concluding remarks

Despite recent progress in estimating global international[7,8,35,36] and internal migration[9,37], global migration datasets often suffer from poor spatial and/or temporal resolutions. Our annual gridded net migration data cover the entire globe, allowing the analysis of both local and regional net migration patterns at various geospatial scales. Using these data, we have quantified the magnitude and impact of net migration over the past 20 years in three administrative levels and in socioclimatic zones. Our analysis highlights the importance of considering the spatial scale when analysing migration patterns. We showed that global net migration patterns depend strongly on the scale of analysis, both for the magnitude of migration and also for the trend (Fig. 2). Within the study period of 2000–2019, net global urban migration has been predominantly positive while net rural migration has been negative (Fig. 3m), aligning with previous urbanization literature[38]. Further, we showed that the volume and impact of migration are positively associated with the level of human development (positive net migration often co-occurs with high HDI) and this association is stronger than that of climate and migration (Fig. 6).

Previous studies argue that migration often originates from areas where people have sufficient capacity to move and to use migration as one form of adaptation to unfavourable environmental conditions, often by migrating to urban areas[29,39–41]. Our analysis provides a global quantification of this argument and shows that rural out-migration hotspots were located in socioclimatic conditions with a middle level of aridity and human development (Fig. 6c,d). This corresponds to about 50% of the global urban population who experienced accelerated population growth in the past two decades (Fig. 5). Notably, a relatively large volume of positive urban net migration took place in locations with medium-to-high levels of human development, indicating higher capacity to cope with additional pressures (Fig. 6a,b).

Despite accelerating urbanization over the past 20 years, positive net migration has increased population growth and helped to slow down population decline in many rural areas (Figs. 5 and 6c,d). Rural–rural migration, especially over short distances, is common particularly in smaller countries with few provincial and communal urban centres that attract migrants from other parts of the country[7]. Our results show that areas where migration slowed rural population decline were located in high-income countries (Fig. 6d), and some countries may have reached a point where urbanization has turned to counter-urbanization[34]. On the other hand, the share of people living in urban areas is expected to grow, especially in lower income regions, such as Africa[42]. With the global average temperatures continuing to rise, impoverished urban areas are particularly vulnerable to the impact

of extreme climate events, given lower capacity to manage population growth and to provide basic services and infrastructure[43]. Increased climatic hazards such as heatwaves, water stress, floods and droughts will increase the vulnerability of people living in the outskirts of poor urban agglomerates, which are already the predominant destinations for migrants, particularly in Africa[44]. Notably, such developments are visible in our study. For instance, migration-induced urban population growth over the past two decades was visible around the fast-growing cities of Nigeria, Angola, Kenya and Tanzania, where the capacity to cope with the accelerating population growth was remarkably low (Figs. 5 and 6).

While our study sheds light on global migration trends and patterns at various spatial scales, there are several limitations. First, given that our data derive net migration from the difference between total and natural population change, it is impossible to distinguish between different types of migrants such as refugees, internally displaced persons or economic migrants. Our data mask out important individual migration events and external shocks, such as conflict, altogether by only describing whether an area experienced more in-migration than out-migration, or vice versa, without being able to differentiate the dynamics of migration flows in and/or out of an area[45]. This can be particularly relevant, for instance, in areas under prolonged conflicts in our study period, such as Afghanistan, Syria and Iraq. Further, our analysis used long-term averages of climatic and socioeconomic conditions, thus masking any interannual or intra-annual variation of net migration related to sudden shocks, such as extreme weather events.

It is also important to note that the data collected from national censuses and other databases (Methods; Extended Data Fig. 1) were available at different temporal resolutions. Data for low-income countries were scarcer, while the information from high-income countries was generally more detailed. In some cases, the sources for births and deaths information were also different. To minimize uncertainties, the produced birth and death grids were adjusted so that the subnational sum of births and deaths matched with the collected subnational (administrative level 1) birth and death counts (Supplementary Data). Further, subnational data at administrative level 1 were harmonized with national data; that is, we ensured that the sum of births and deaths at the grid level agree at the national level with United Nations (UN) statistics. We validated the WorldPop data, downscaled births and deaths data as well as the produced net migration data against reported values, mostly at administrative level 2 (finer level than used for creating the gridded datasets). Validations show that the data are in line with reported data and thus indicate good performance of our downscaling method (Supplementary Figs. 1–6 and Supplementary Table 2). To smooth over inaccuracies, we recommend aggregating the data over some spatial or temporal units (for example, administrative/environmental zoning, 5 year period instead of an annual timestep).

Here, we have quantified the magnitude and impacts of net migration over the past two decades. In the light of our results, we emphasize the importance of subnational analysis of migration and, with our data, we provide various future research openings in studying complex human mobilities inside and between administrative boundaries and in other zones—combined with the co-occurrence of extreme weather, disasters, conflicts and migration events, for example. Capturing migration patterns not only between countries but also within countries and socioeconomic and geophysical zonings is essential for policy design, international cooperation and shared responsibility—all essential for managing internal and international migration and other mobilities[29]. Finally, our results underline the necessity of coupling environmental drivers with proxies of human development. Internal and international migration are expected to increase in unprecedentedly complex and rapidly changing socioclimatic conditions[46] and thus, increasing the empirical understanding of the global patterns of migration and its multifaceted drivers is very important[47].

## Methods

The overall workflow is illustrated in Fig. 1, while a more detailed methodological explanation is given below.

### Data processing for harmonized subnational data

We first collected national and subnational data for 1990–2019. For births, we used two global databases, namely STATcompiler[48] and Eurostat[49], as well as national census data. Altogether we found subnational (administrative level 1) data for 163 countries which were divided altogether for 2,555 administrative units (Extended Data Fig. 1). For deaths, we used two databases, namely OECD regional stats[50] and Eurostat[51], as well as national census data. For deaths, we found subnational data for 124 countries and 2,067 administrative units. For both data (births and deaths) and for each country, we evaluated the best-available data source (by the number of available years and missing data entries, if multiple data sources were available). The datasets used are described in the Supplementary Data and origin of data for each country illustrated in Extended Data Fig. 1, which also shows the administrative units used in STATcompiler[52], Eurostat[53] and OECD[54].

As data were available either as a crude birth or death rate or as a number of births or deaths, we needed to harmonize the data so that all observations would be as a rate (births or deaths per 1,000 people). For this, we aggregated population data for each administrative unit and for each year, using WorldPop[55] gridded 1 km data for 2000–2019 and HYDE 3.2 (ref. 56) for 1990–2000. HYDE data were bias-corrected with delta change method[57] to be consistent with WorldPop data. Although WorldPop data were available also for the year 2020 at the time of data processing, our analysis required computing population change that could not be done for the 2020 WorldPop data and thus, final data are limited to 2019. While our final dataset covers years 2000–2019, collecting data for the previous decade made interpolations between years possible.

Once all data were transformed to rates, we filled missing data entries. Some data entries were missing due to changes in administrative units and temporal gaps in used datasets. For changes in administrative boundaries, we used Administrative Divisions of Countries (Statoids) database[58]. We processed the missing data according to following types (see list of subnational areas in each category in the Supplementary Data):

(1) Combine (number of cases in birth dataset ($n_{births} = 11$; $n_{deaths} = 11$). In subnational units with low population (typically <50,000)—with available data for the number of births or deaths—the birth or death rates ended up differing substantially from the surrounding administrative units. In these cases, we 'combined' the unit with a neighbouring larger subnational unit so that the rate was calculated for both, using the combined number of births or deaths and population—that is, they ended having the same rate. For example, we combined Ulaanbaatar (Mongolia) with the surrounding Töv province, as rates in Ulaanbaatar were not realistic (probably because of how the population in the capital is defined).

(2) Split ($n_{births} = 15$; $n_{deaths} = 14$). In cases when a larger administrative unit was split to two or more subnational units during the extended study period and parts of the data were given only for the larger unit, we estimated birth or death rates before the split using the birth or death rate of the first timestep when these were split to separate units. We did this by calculating the combined birth or death rate, using subnational population and then used this to calculate the ratio between combined birth or death rate and each individual administrative unit birth or death rate. These were then multiplied with the 'combined' birth or death rate for years before the split. This way, we were able to estimate the birth or death rate for all individual administrative units for the time in which they were still one unit.

For example, Panamá subnational unit was split in 2014 into two units: Panamá and Panamá Oeste. Birth rate data exist for combined Panamá over 2001–2013 and then for both split units from the year 2014 onwards. We thus first calculated the combined birth rate (in this case) for year 2014 using subnational population:

$$\text{birthRate}_{\text{comb}_{2014}}$$
$$= \frac{\left(\text{pop}_{\text{PanamaO}_{2014}} \times \text{birthRate}_{\text{PanamaO}_{2014}}\right) + \left(\text{pop}_{\text{Panama}_{2014}} \times \text{birthRate}_{\text{Panama}_{2014}}\right)}{\left(\text{pop}_{\text{PanamaO}_{2014}} + \text{pop}_{\text{Panama}_{2014}}\right)} \quad (1)$$

This $\text{birthRate}_{\text{comb}_{2014}}$ was then used to estimate the rates for each subnational unit separately:

$$\text{ratio}_{\text{PanamaO}} = \frac{\text{birthRate}_{\text{PanamaO}_{2014}}}{\text{birthRate}_{\text{comb}_{2014}}} \quad (2)$$

$$\text{ratio}_{\text{Panama}} = \frac{\text{birthRate}_{\text{Panama}_{2014}}}{\text{birthRate}_{\text{comb}_{2014}}} \quad (3)$$

and these were finally used to estimate the birth rate for each subnational unit for pre-2014 period (example for year 2013 given below):

$$\text{birthRate}_{\text{PanamaO}_{2013}} = \text{ratio}_{\text{PanamaO}} \times \text{birthRate}_{\text{Panama(combined)}_{2013}} \quad (4)$$

$$\text{birthRate}_{\text{Panama}_{2013}} = \text{ratio}_{\text{Panama}} \times \text{birthRate}_{\text{Panama(combined)}_{2013}} \quad (5)$$

where birthRate_Panama(combined)_2013 is the reported birth rate for the combined administrative unit for year 2013 (before the split).

(3) Part of the data missing ($n_{\text{births}} = 31$; $n_{\text{deaths}} = 19$). In cases of gaps in the reported data, we used the neighbouring area with the least difference in births or deaths as the scaling neighbour to scale the missing data. For this, we used the data from the closest year to the missing data year, for which data existed for both administrative units. We then calculated the ratio between these administrative units and used this ratio to estimate the missing data:

$$\text{data}_{\text{admMissing}_{\text{missingYear}}} = \frac{\text{data}_{\text{admMissing}_{\text{availableYear}}}}{\text{data}_{\text{admExisting}_{\text{availableYear}}}} \times \text{data}_{\text{admExisting}_{\text{missingYear}}} \quad (6)$$

(4) No data at all ($n_{\text{births}} = 24$; $n_{\text{deaths}} = 29$). For some subnational units (often remote islands or other remote areas with very low population), no data existed. In this case, we used national average values.

Once we had filled the missing data entries, we interpolated and extrapolated the years for which there were no data available for any subnational unit, using interpolation and harmonization methods adapted from ref. 59. Shortly, we first interpolated missing national values between the years with available data using linear interpolation. Then, we extrapolated missing data points either before the first reported year or after the last one. For extrapolation, we used macroregional ($n = 12$; see ref. 59) trends that were calculated using population-weighted national birth and death rates of those countries with full data coverage over the extended study period (1990–2019).

We then used the interpolated and extrapolated subnational birth and death rates to calculate population-weighted national average birth and death rates for each country and year. These rates were then compared with reported national values (compiled from World Bank and UN databases), resulting in an annual ratio between the reported death and birth rates and the ones calculated from subnational

units. When we then used this national ratio to correct the bias of the subnational-level birth and death rates, we ensured that our data matched with reported data at the national level.

As the final result, we attained gap-filled harmonized full time series for all subnational areas for which data were available as well as for those countries for which no subnational data were found (Extended Data Fig. 1). We produced a combined dataset (both gridded and polygon formats) for both birth and death rates, which consisted of subnational values for those areas where it was available and national values for others.

### Downscaling to gridded level

We used multiple linear regression together with area-to-point kriging (ATPK)[60,61] to downscale the subnational birth and death rates to 5 arcmin (~10 km at the equator) resolution. The downscaling was done so that the average birth and death rates (total births and deaths of an area) were maintained at the administrative unit (subnational or national, depending on the source data) scale[62]. For the regression model, we used 'relative population density' (for urbanization proxy) and 'subnational human development index' data as independent variables in downscaling both births and deaths. Income and mean years of schooling, both being indicators of HDI, have been shown to correlate well with babies per woman at national scale[63] and ref. 5 used population density as a proxy for estimating the difference in natural population increase between urban and rural areas. For births, these were used together with gridded data of 'share of women of reproductive age (15–49 years) of total population' and for deaths with gridded data of 'ratio of average age and life expectancy'. The 'share of women of reproductive age (15–49 years) of total population' is also used by ref. 64 to downscale births to gridded scale—although they did it only for one year and selected countries. For the independent variables used for downscaling, we used annual datasets for 2000–2019. These data were produced for this analysis by using openly available global datasets. Data preparations for the independent variables are described below, followed by the downscaling method in details.

**Data preparation of independent variables for downscaling.** Human development index. We used tabulated subnational HDI data from ref. 65 to create a gridded and gap-filled gridded subnational HDI dataset for each year over 2000–2019, at 5 arcmin resolution. We used a similar method to fill missing years and rasterize the data as ref. 59 use in their gridded HDI dataset.

**Relative population density.** Rather than using global population density or night light data as the proxy of urbanization rate, we used population density scaled between 0 and 1 for each country. Using the WorldPop dataset[55], we first extracted the 5th and 95th percentiles of population density by country across all years from 2000 to 2020, after the lowest population density cells (population density < 1 person per km²) were omitted. We then scaled population densities within each country and year between 0 and 1, such that:

$$\text{popdScaled} = \frac{\text{popd} - \text{popd5th}}{\text{popd95th} - \text{popd5th}} \quad (7)$$

$$\text{popdScaled} < 0 = 0$$

$$\text{popdScaled} > 1 = 1$$

where popdScaled is the scaled population density for a year, popd is the unscaled population density for that year, popd95th is the 95th percentile population density for the country in question (using all years) and popd5th is the 5th percentile population density (using all years).

The scaled population density then indicates the degree of urbanization an area has undergone relative to the maximum state of urbanization experienced by that country across 2000–2020.

**Share of women of reproductive age of total population.** We used WorldPop 'Age and sex structures' dataset[66] to count the number of women of reproductive age (15–49 years). The data provide annual estimates globally for each 30 arcsec grid cell for the "[…] total number of people per grid square broken down by sex and age groupings (including 0–1 and by 5-year up to 80+)" (ref. 67). We aggregated these data to 5 arcmin resolution and divided this with the aggregated (from 30 arcsec to 5 arcmin) total population count of each grid cell using WorldPop 'Population counts' data[55]. This resulted in the share of women of reproductive age for each grid cell at 5 arcmin resolution. We repeated this for years 2000, 2005, 2010, 2015 and 2020 and interpolated the missing years to get annual gridded data for 2000–2020 (see map of the last timestep in Fig. 1e).

**Ratio of average age and life expectancy.** To first calculate the average age of population in each grid cell, we used the same WorldPop 'Age and sex structures' dataset[66] that was used to compute the 'share of women of reproductive age (15–49 years) of total population' (see above). The data include the number of males and females by age group and thus, it allowed us to estimate the average age (for example, for the age group 5–9 years, we used an average age of 7 years). However, the dataset does not specify the average age for the age group 80+ and, thus, we needed to use UN population prospects national-level data to estimate it for each country and year and separately for females and males. From the national-level data, we first estimated the average age of that age group (separately for females and males) and then combined that with the grid cell level data—that is, assuming that in each grid cell of a country in question, the share of population over 80 years old had an average age equal to national average. As a result, we reached an estimate of the average age for each 5 arcmin grid cell. We then used the life expectancy component of subnational HDI data from ref. 65. To fill missing years and rasterize the data, we used a method similar to the one ref. 59 use in their gridded HDI dataset. Finally, we took the ratio of these two datasets, resulting in a ratio of average age and life expectancy—the percentage of how much of their expected length of life has an average person lived—for each year over our study period 2000–2019.

**Downscaling methods.** To downscale the collated census data from the subnational administrative boundary scale to 5 arcmin gridded scale, we fitted multiple linear regression models of death and birth rates against global independent variables introduced above. Separate regression models were fitted for four income groups, where grouping was based on their relative income band[68]: low, lower middle, upper middle or high income. We first applied these regression models at the subnational administrative boundary scale to assess model performance and to calculate residuals (differences between model estimates and reported data) for each subnational administrative area. We then applied the regression models at the 5 arcmin scale and adjusted the estimates using an ATPK for the residuals calculated at the subnational administrative boundary scale. Finally, the downscaled estimates were adjusted to ensure that the total number of births and deaths matched the original data when aggregated to the subnational administration boundary scale[62]. Below, each step is presented in more detail.

**Linear regression models.** On the basis of the dataset from the World Bank[68], we split countries into four income bands, marking them as low, lower middle, upper middle or high income. For each income bracket, we fitted a different regression model to capture the differing relationships between birth or death rates and independent variables.

Both birth and death rate models used the scaled population density and HDI data as predictor variables, with birth rate models additionally including the 'share of women of reproductive age (15–49 years) of total population' variable, whilst death rate models additionally included the 'ratio of average age and life expectancy' variable. The predictor variables were aggregated at the subnational administrative boundary scale by taking average values, which were then used for regression against either birth rates or death rates across all years (2000–2019).

We applied these regression models to predict birth and death rates for each grid cell, aggregated the cell-wise values at the subnational scale by weighting them with population and then assessed model performance and calculated residuals against the collated census data. We found that the models predicted birth and death rates well, with the coefficient of determinations being 0.74 and 0.60, respectively (Supplementary Table 1).

**ATPK of the residuals and final population-weighted adjustment.** Having calculated the residuals between modelled estimates of birth and death rates at the subnational scale and the collated census data, we performed an ATPK procedure to distribute these residuals across a 5 arcmin point mesh following the method described in ref. 69. These grid-scale distributions of the residuals were then added to a grid-scale application of the regression models to give a first approximation for the downscaled datasets of both birth and death rates.

Finally, we adjusted the downscaled approximations to ensure that the total number of births or deaths represented in the downscaled dataset matched the collated census data at the subnational scale by applying equation (8) to each grid cell:

$$R_{i\text{ adjusted}} = R_i \left( \frac{C_n \sum_{j=1}^{j=n} p_j}{\sum_{j=1}^{j=n} R_j p_j} \right) \tag{8}$$

where $R_i$ adjusted is the adjusted rate (birth or death) in the grid cell, $R_i$ is the approximated rate in the grid cell, $j = 1…n$ are the grid cells that fall within each subnational administrative boundary, $p_j$ is the population within grid cell $j$ taken from WorldPop and $C_n$ is the census data birth or death rate for the subnational boundary containing cells $j = 1…n$.

**Net migration data**
The downscaled gridded births and deaths data (Fig. 1g,h) were then used to calculate natural population change (deaths subtracted from births) (Fig. 1i). Together with annual global gridded population count for 2000–2020[55] (Fig. 1j), the natural population change data allowed us to calculate net migration for each year and grid cell (Fig. 1k) as 'natural population change' subtracted from 'reported population change'.

**Validation of data**
We validated all downscaled datasets (WorldPop data and gridded births, deaths and net migration) with subnational and/or national observations. Subnational observations at administrative level 2 were collected from OECD[50,70], Eurostat[49,51,71], national statistical service of South Korea[72] and from ref. 73. National observations were collected from the UN[74]. Here, it should be noted that the subnational validation data for population, births and deaths were collected at administrative level 2 (communal) whereas level 1 (provincial) data were used for producing and harmonizing the gridded datasets. Level 1 (provincial) data were used in validating the net migration estimates.

**WorldPop.** WorldPop data were validated against OECD data[70] which reports population counts for 1,818 subnational units globally. WorldPop data were first aggregated to the subnational units and then

compared with OECD data through correlation analysis. Validation results are presented in detail in Supplementary Fig. 1 and Supplementary Table 2.

**Downscaled births and deaths.** Downscaled births and deaths were validated against Eurostat subnational data ($n = 1,504$) and OECD subnational data ($n = 1,883$), respectively. Gridded birth and death rates were first aggregated to the respective subnational units and then compared with the observed values. Validation results are provided in the Supplementary Fig. 2 and Supplementary Table 2.

**Net migration.** For validating net migration data, we used national-level data provided by the UN[74] and subnational observations collected from ref. [73] for the United States (counties; $n = 3,056$), national statistical service of South Korea (provinces; $n = 16$) (ref. [72]) and Eurostat for Europe (districts; $n = 1,522$) (ref. [71]). Gridded net migration data were first aggregated to the national/subnational units and then compared with the observations in the respective resolution. Subnational validation was conducted for both the total net migration count as well as the rate (net migration per population). Total net migration for a unit was first obtained by aggregating (zonal sum) the gridded net migration estimates over a subnational unit. Annual rate of net migration per population for a subnational unit could then be calculated by dividing both the aggregate sum of net migration and the observed net migration by the aggregate sum of population count (here, World-Pop data were used) of the respective subnational unit. Additionally, cumulative sums of net migration were calculated for both the estimated (here-produced) and observed net migration data. Cumulative sums were calculated for years 2000–2005, 2006–2010, 2011–2015, 2015–2020 (5 year sums) and for years 2000–2010 and 2011–2020 (10 year sums). Rates for the accumulated net migration counts were calculated by dividing the accumulated sum by the respective 5 or 10 year population average in the given subnational unit. Validation results and country-specific details are provided in Supplementary Figs. 3–6 and Supplementary Table 2.

## Urban extent data

Existing mappings of global urban areas include exercises like the one mapping global human settlements and urban centres by using a 'degree of urbanization' approach that divides human settlements into rural, semi-urban and urban areas on the basis of population size, population and the density of built-up areas. However, these data cover only a few years[75]. A recent urban extent dataset developed by using night light data also covers only one timestep[76], while the most up-to-date time-series data of global urban extent cover nearly 100 years (2010–2100) (ref. [77]), hence missing a decade from our study period.

Thus, we created urban extent data for each country for each year by using gridded relative population density (see above) and the share of urban population of total population for each country and year. The tabulated annual share of urban population in each country was acquired from the UN World Urbanization Prospects[42]. By going through each country cell by cell, the share of urban population was used to find a population density threshold where urban population turns to rural. Here, it was assumed that all urban area cells were more densely populated than all rural area cells. In other words, the calculation propagated in descending order of population density so that the sparsest-populated cell within urban extent was assumed to be more densely populated than the densest-populated cell not within the urban extent.

This was done by first calculating the total and urban population for each country:

$$totPop = sum(pop) \tag{9}$$

$$urbanPop = totalPop \times shr_{urbanPop} \tag{10}$$

Then, a threshold where urban population turns to rural was defined by calculating the cumulative population count. Starting from the cells with highest relative population density, population count in each grid cell in each country was summed until the cumulative count met the urban population defined above (urbanPop). All cells in the cumulative count were assigned value 1 (urban) and the rest were assigned value 0 (rural).

## Zonal statistics: administrative areas and socioclimatic bins

For the three administrative levels, we used the Global Administrative Areas (GADM) levels 0, 1 and 2 for national, provincial and communal areas, respectively.

All zonal statistics in the study were done with the *zonal* tool in *terra* package (v.1.5-12) in R[78]. Socioclimatic bins were created by using global gridded data of aridity (global aridity index[79]), human development (human development index; see above) and population counts for 2000–2019 (ref. [55]). For the climatic factor, aridity index—a ratio of potential evaporation to precipitation—was chosen to capture the different atmospheric and land surface processes shaping terrestrial dryness. Aridity index has been used to assess desertification under climate change[80]. Here, the zonal analyses were conducted by using a long-term estimate of net migration (accumulated sum over the study period 2000–2019), which is why the impacts of short-term events, such as natural disasters, are not visible in the analysis.

Our binning divides global inhabited areas into 100 socioclimatologically analogous zones, which have similar human development and climatic conditions. The binning was conducted in two steps. First, we divided all considered grid cells into ten population-weighted quantiles on the basis of HDI. After that, each HDI quantile was again divided into ten population-weighted quantiles on the basis of aridity. All values in the first aridity quantile of the first HDI quantile would fall under the first bin (bin no. 0), while values in the second aridity quantile of the first HDI quantile would go in the second bin (bin no. 1). The division of bins is illustrated in Extended Data Fig. 7. This division ensured that each bin incorporates ~1% of the global population. The binning represented as a heatmap could then be transformed into a map representation (Extended Data Fig. 8c). Urban and rural areas in each bin could be extracted by using the urban extent data (see above). Global maps and description of the preprocessing of both human development, aridity index and socioclimatic bins are available in Extended Data Fig. 8.

## Impact calculations

Impact of total, rural and urban migration on respective populations was calculated for each administrative unit, as well as for each socioclimatic bin. First, migration data and population counts were aggregated for each spatial unit for each year (2000–2019) by using zonal sum. Then, total population change between 2019 and 2000 and accumulated net migration were calculated for each unit:

$$popChange = popChange_{2019} - popChange_{2000} \tag{11}$$

$$netMgrSum = \sum_{year=2001}^{2019} netMgr_{year} \tag{12}$$

Population change without migration (natural population change) could then be calculated by subtracting cumulative net migration from total population change:

$$popChange\_woMgr = popChange - netMgrSum \tag{13}$$

The impact of migration on population was deducted by comparing population change to cumulative net migration by using the following criteria:

| (1) popChange_woMgr > 0 & netMgrSum > 0 | Migration increases natural population growth |
|---|---|
| (2) popChange < 0 & netMgrSum > 0 | Migration slows down declining population |
| (3) popChange_woMgr < 0 & popChange > 0 | Migration turns declining population to growth |
| (4) popChange_woMgr < 0 & netMgrSum < 0 | Migration increases natural population decline |
| (5) popChange > 0 & netMgrSum < 0 | Migration slows down growing population |
| (6) popChange_woMgr > 0 & popChange < 0 | Migration turns growing population to decline |

### Direction of migration

The direction of migration was studied in rural and urban areas at three administrative levels. Here, accumulated rural and urban net migration over 2000–2019 were calculated and compared to define the direction of migration according to the following criteria:

| (1) UrbanMgr > 0 & RuralMgr > 0 | Urban and rural net migration positive: net-receiving area |
|---|---|
| (2) UrbanMgr < 0 & RuralMgr < 0 | Urban and rural net migration negative: net-sending area |
| (3) UrbanMgr < 0 & RuralMgr > 0 | Urban net migration negative and rural net migration positive: rural pull–urban push |
| (4) UrbanMgr > 0 & RuralMgr < 0 | Urban net migration positive and rural net migration negative: urban pull–rural push |
| (5) UrbanMgr == 0 & RuralMgr == 0 | Urban and rural net migration are zero |

### Reporting summary

Further information on research design is available in the Nature Portfolio Reporting Summary linked to this article.

### Data availability

All the data used in this study are publicly available. The resultant datasets are available at the following open-access repository https://doi.org/10.5281/zenodo.7997134, including the following datasets: annual global net migration rates at grid scale as a multiband GeoTIFF with 5 arcmin resolution for 2000–2019; annual global net migration rates for adm0, adm1 and adm2 levels as polygon layers (gpkg-files); and annual global birth and death rates as multiband GeoTIFFs with 5 arcmin resolution for 2000–2019. Data are visualized in online net migration explorer at https://wdrg.aalto.fi/global-net-migration-explorer/. Data underlying the web application are available in the repository with all other data.

### Code availability

The analysis was performed using RStudio (R v.4.1.2). The code is available at https://github.com/mattikummu/global_net_migration.git. The online net migration explorer was conducted using RStudio (R v.4.2.2). The code is available at https://github.com/vvirkki/net-migration-explorer.git.

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

## Acknowledgements

This study was funded by Maa- ja vesitekniikan tuki ry, the European Research Council under the European Union's Horizon 2020 research and innovation programme (SOS.aquaterra project; grant no. 819202), the European Research Council under the European Union's Horizon 2020 research and innovation programme (POPCLIMA project, grant no. 101002973), the Aalto University School of Engineering, the Academy of Finland (TREFORM project; grant no. 339834), the Academy of Finland (WATVUL project; grant no. 317320) and the National Science Foundation of China funding research fund for International Young Scientists (grant no. 41950410572). The funders had no role in study design, data collection and analysis, decision to publish or preparation of the manuscript. We would like to thank M. Jalava from Aalto University for help with issues related to computations and setting up a server for the web application.

## Author contributions

V.N. and M. Kummu designed the research. Data collection and processing was led by M. Kummu and M. Kosonen with help from A.H., V.V., M.H., M. Kallio, P.K. and V.N. V.N. and M. Kummu performed the analysis with help from A.H. V.V. created the online net migration explorer. V.N. and M. Kummu created the illustrations. All authors discussed the methods and results and helped shape the research and analysis. V.N. and M. Kummu took the lead in writing the manuscript with important contributions from all authors.

## Funding

## Competing interests

The authors declare no competing interests.

## Additional information

**Extended data** is available for this paper at https://doi.org/10.1038/s41562-023-01689-4.

**Correspondence and requests for materials** should be addressed to Venla Niva or Matti Kummu.

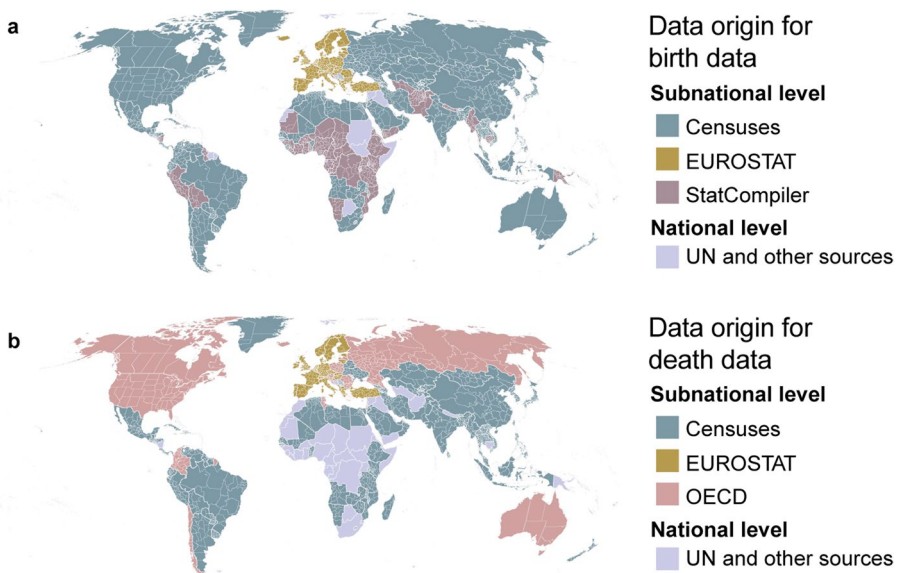

**Extended Data Fig. 1 | Origin of datasets used.** Origin of the datasets used for a) birth rates and b) death rates. See Methods for more details. Sources for the data are provided in Supplementary Data Table. The geospatial files used in the maps are from The DHS Program[52], Eurostat[53], OECD[54] and GADM (www.gadm.org).

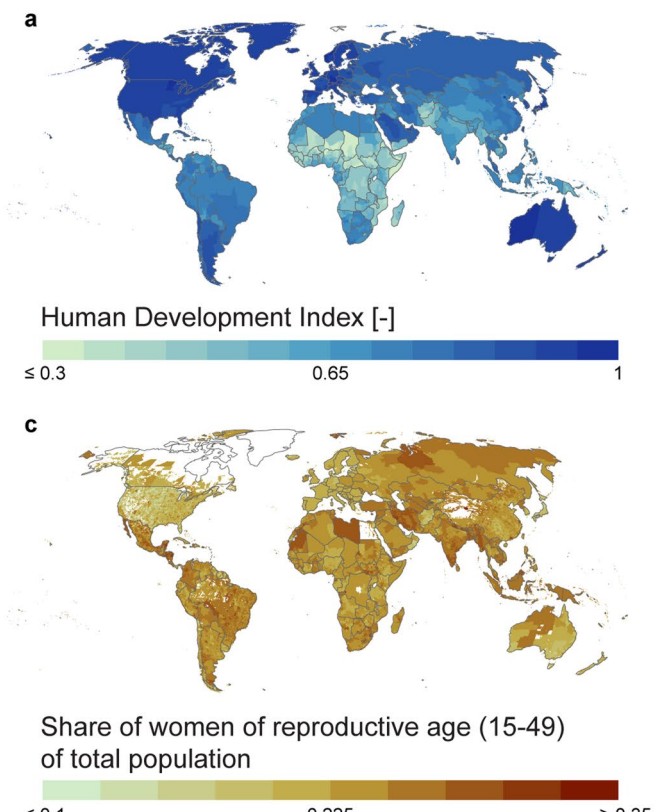

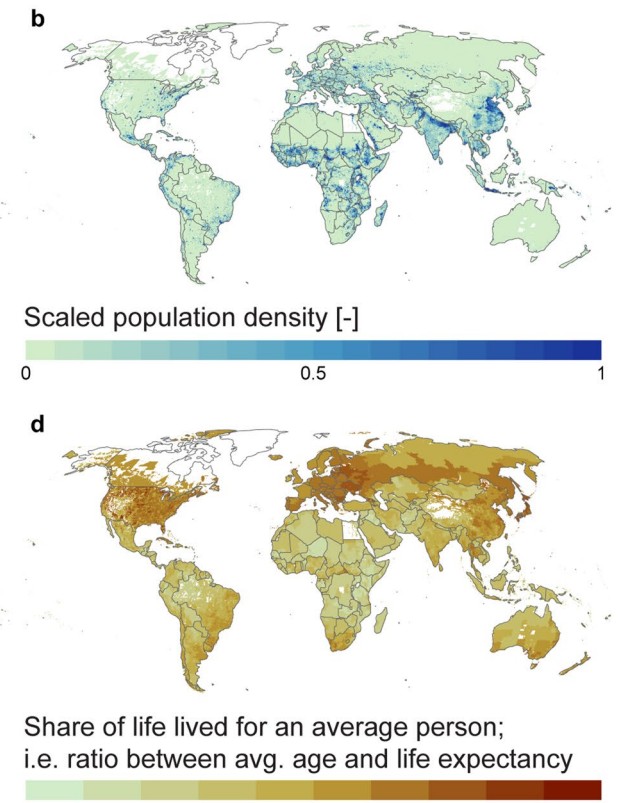

**Extended Data Fig. 2 | Input variables used for downscaling birth and death rates.** (a) Human Development Index, (b) population density, scaled between 0–1, (c) share of women of reproductive age (15–49) of total population and (d) share of life lived for an average person. Each dataset includes annual values for each year over 2000–2019 – shown here are the data for the last timestep, that is year 2019.

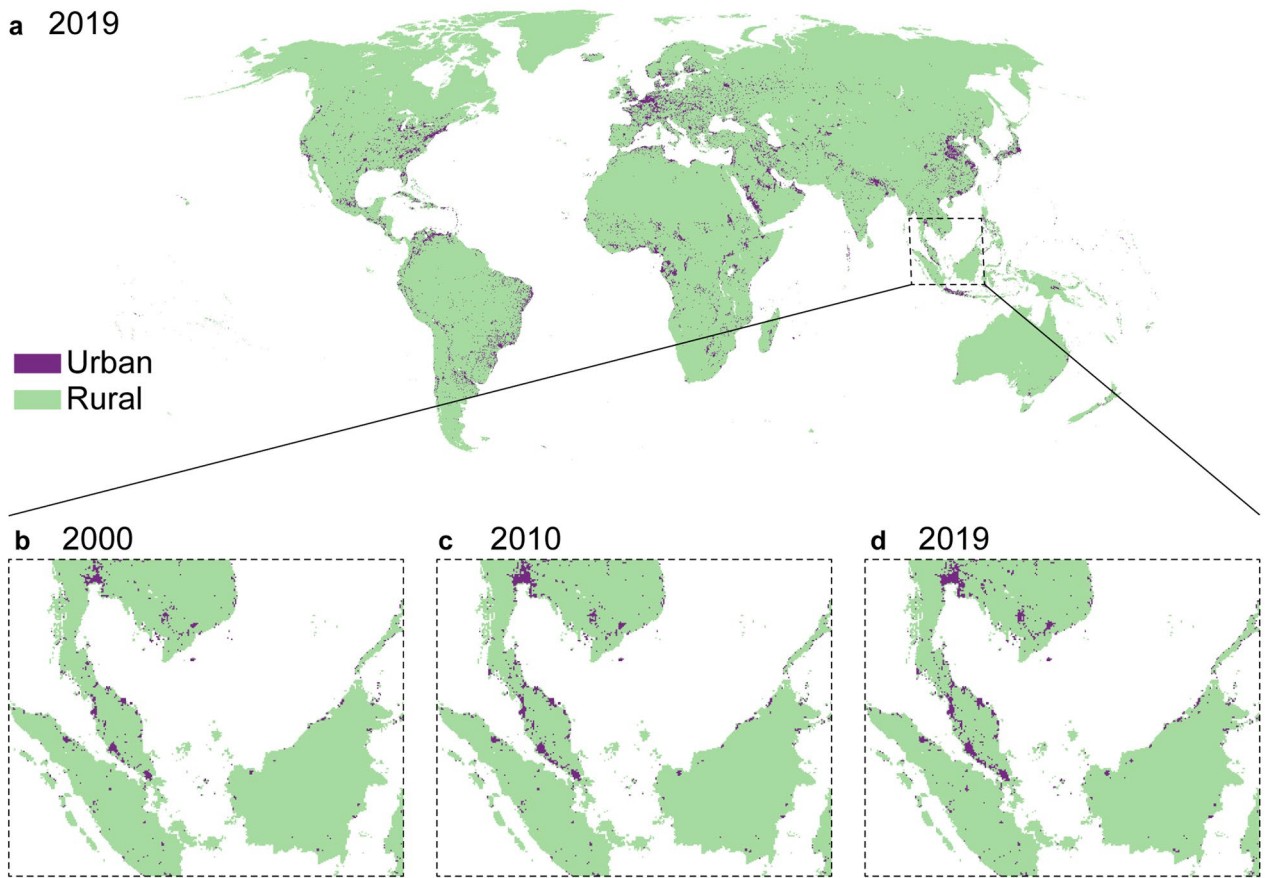

**Extended Data Fig. 3 | Urban and rural areas.** Example of (a) global urban areas in 2019 and in three timesteps (b: 2000, c: 2010, d: 2019) in the South China Sea around the Malaysian peninsula.

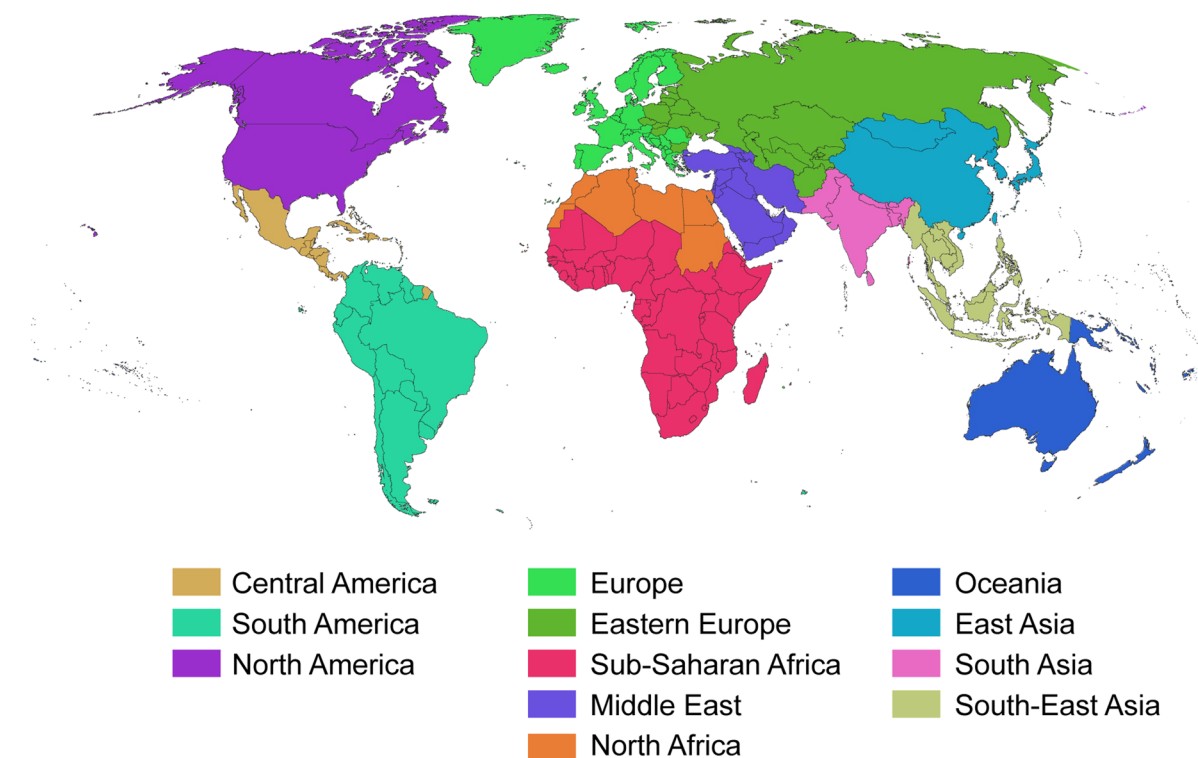

**Extended Data Fig. 4 | Regional division.** Regional division based on the UN country grouping.

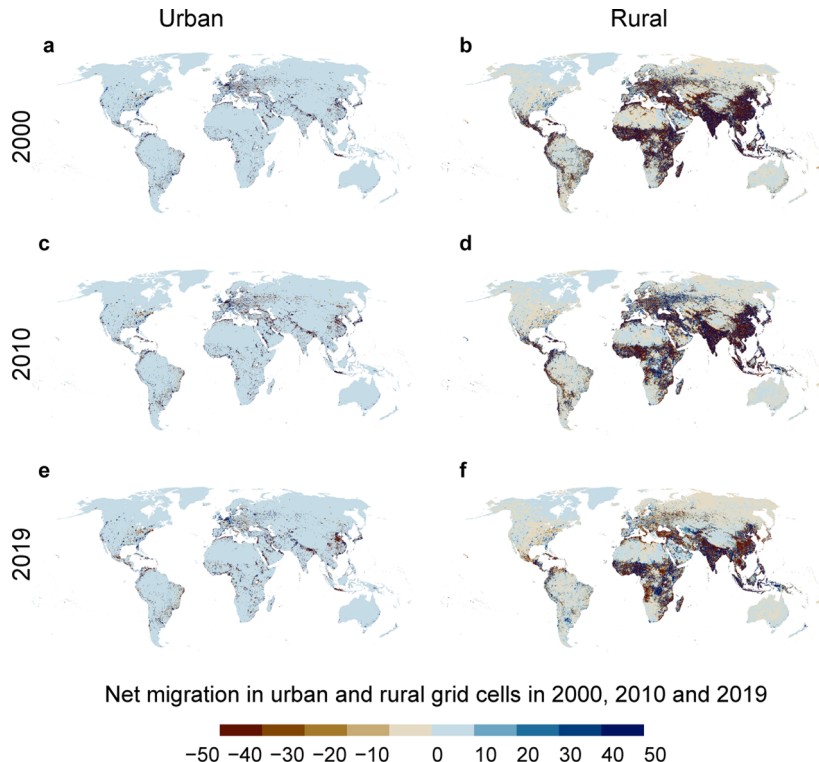

Net migration in urban and rural grid cells in 2000, 2010 and 2019

**Extended Data Fig. 5 | Net migration in rural and urban areas.** Net migration in urban and rural grid cells in three timesteps: 2000 (a,b), 2010 (c,d) and 2019 (e,f).

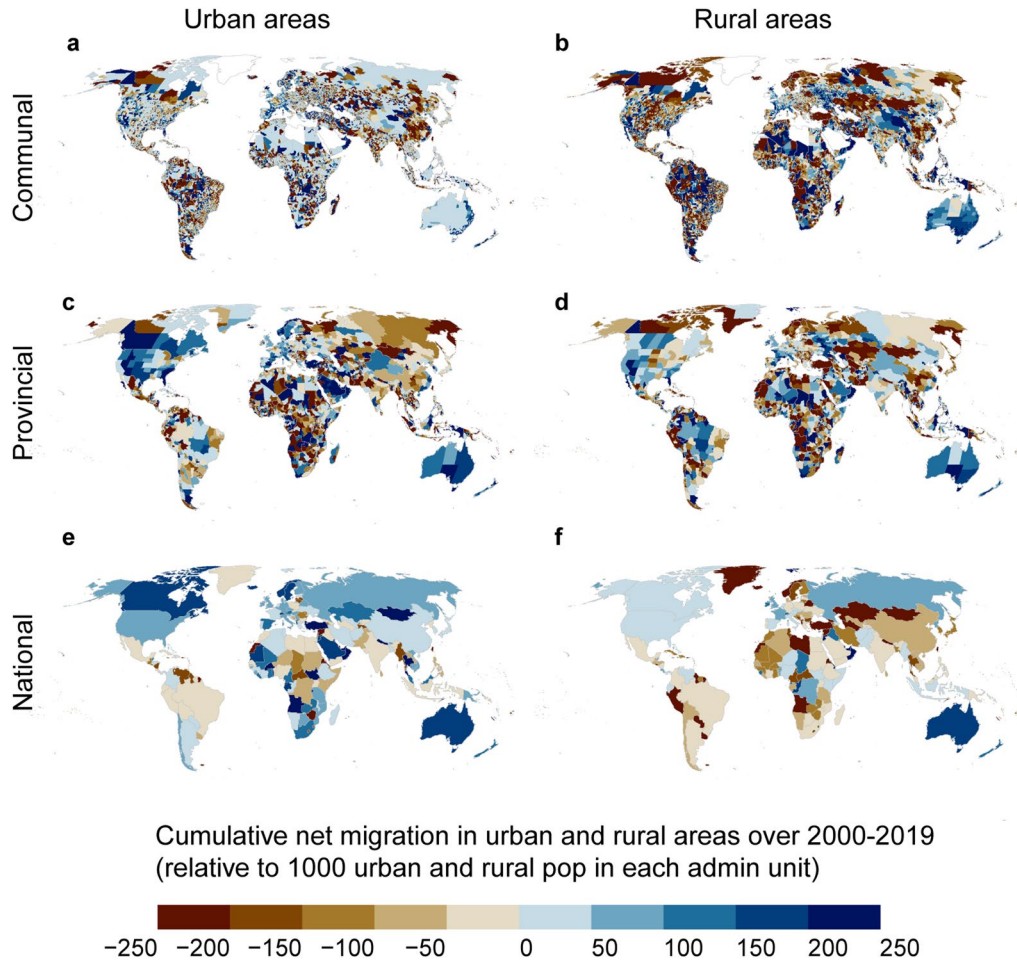

Urban areas — Rural areas

Cumulative net migration in urban and rural areas over 2000-2019
(relative to 1000 urban and rural pop in each admin unit)

−250 −200 −150 −100 −50 0 50 100 150 200 250

**Extended Data Fig. 6 | Cumulative net migration in urban and rural areas.** Cumulative net migration (per 1000 urban and rural inhabitants in each administrative unit) in urban and rural areas over 2000–2019. Three administrative levels are shown: a-b: communal (admin 2 level), c-d: provincial (admin 1 level) and e-f: national (admin 0 level).

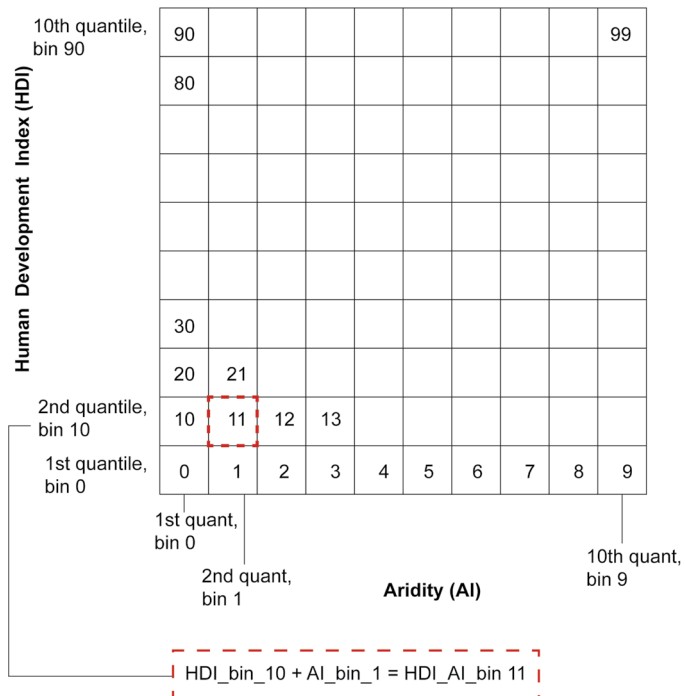

**Extended Data Fig. 7 | Schematic illustration of creating the socioeconomic bins.** Illustration of how the Human Development Index (HDI) and Aridity (AI) were combined to create socioeconomic bins. Each bin includes ca 1% of global population.

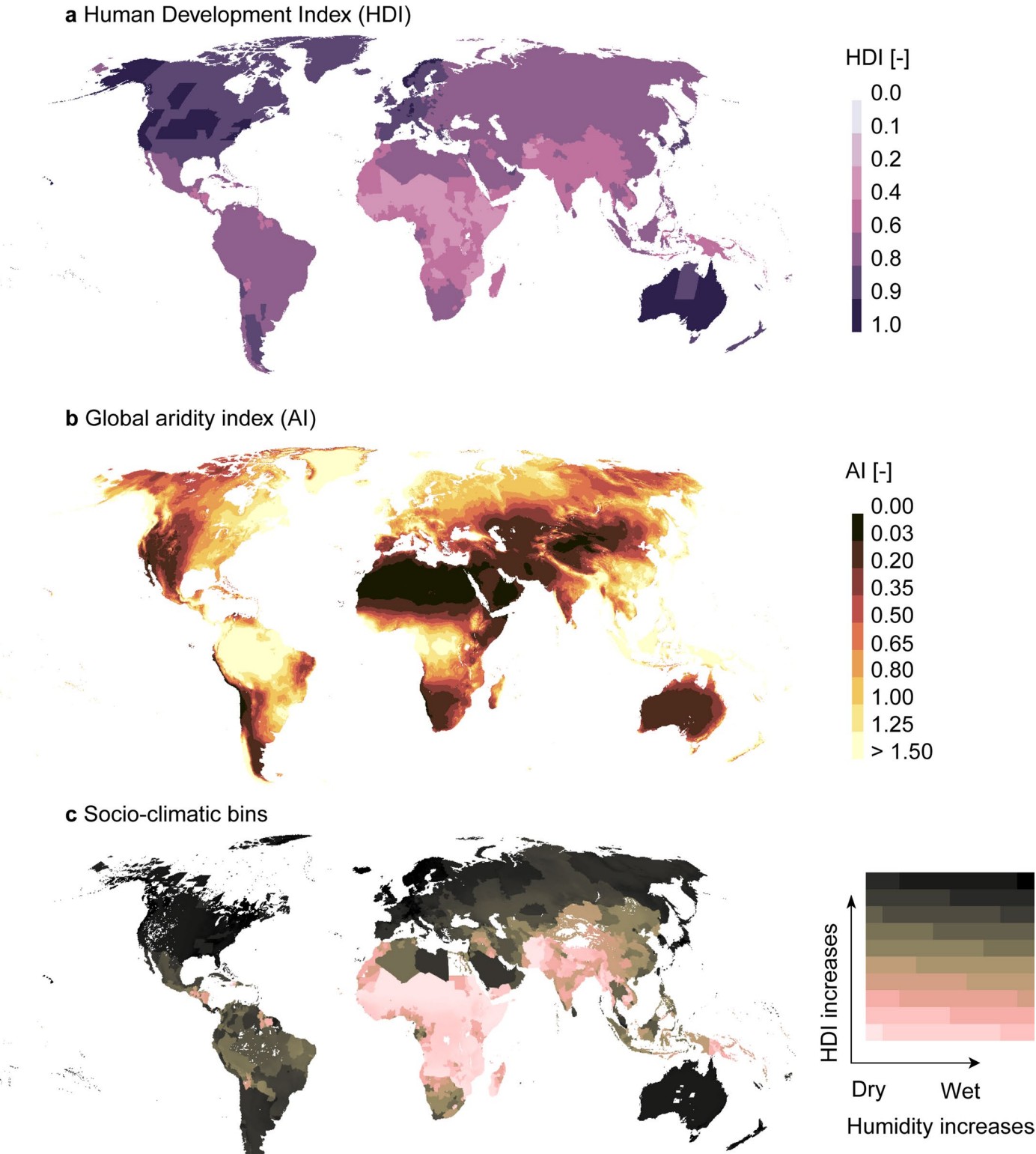

**Extended Data Fig. 8 | Socioclimatic bins with the input data for them.** Global maps of (a) human development index, (b) global aridity index and (c) socioclimatic bins. Each bin includes ca 1% of global population. See Methods for more details.

# Reporting Summary

## Statistics

For all statistical analyses, confirm that the following items are present in the figure legend, table legend, main text, or Methods section.

| n/a | Confirmed | |
|---|---|---|
| ☐ | ☒ | The exact sample size (*n*) for each experimental group/condition, given as a discrete number and unit of measurement |
| ☒ | ☐ | A statement on whether measurements were taken from distinct samples or whether the same sample was measured repeatedly |
| ☐ | ☒ | The statistical test(s) used AND whether they are one- or two-sided *Only common tests should be described solely by name; describe more complex techniques in the Methods section.* |
| ☒ | ☐ | A description of all covariates tested |
| ☐ | ☒ | A description of any assumptions or corrections, such as tests of normality and adjustment for multiple comparisons |
| ☒ | ☐ | A full description of the statistical parameters including central tendency (e.g. means) or other basic estimates (e.g. regression coefficient) AND variation (e.g. standard deviation) or associated estimates of uncertainty (e.g. confidence intervals) |
| ☐ | ☒ | For null hypothesis testing, the test statistic (e.g. $F$, $t$, $r$) with confidence intervals, effect sizes, degrees of freedom and *P* value noted *Give P values as exact values whenever suitable.* |
| ☒ | ☐ | For Bayesian analysis, information on the choice of priors and Markov chain Monte Carlo settings |
| ☒ | ☐ | For hierarchical and complex designs, identification of the appropriate level for tests and full reporting of outcomes |
| ☒ | ☐ | Estimates of effect sizes (e.g. Cohen's *d*, Pearson's *r*), indicating how they were calculated |

*Our web collection on statistics for biologists contains articles on many of the points above.*

## Software and code

Policy information about availability of computer code

| Data collection | The data used in the analysis was collected manually from national censuses, demographic databases (StatCompiler and EuroStat) and other open data repositories. No software were used for data collection. |
|---|---|
| Data analysis | The analysis conducted in this paper was conducted with R (version 4.1.2) using R-Studio (RStudio 2021.09.0+351 "Ghost Orchid" Release). All R-scripts created for data processing, analysis and visualisation will be made openly available in GitHub upon publication. |

For manuscripts utilizing custom algorithms or software that are central to the research but not yet described in published literature, software must be made available to editors and reviewers. We strongly encourage code deposition in a community repository (e.g. GitHub). See the Nature Portfolio guidelines for submitting code & software for further information.

## Data

Policy information about availability of data

All manuscripts must include a data availability statement. This statement should provide the following information, where applicable:
- Accession codes, unique identifiers, or web links for publicly available datasets
- A description of any restrictions on data availability
- For clinical datasets or third party data, please ensure that the statement adheres to our policy

All the data used in this study are publicly available. The resulted datasets are available at the following open-access repository: http://doi.org/10.5281/zenodo.7997134, including the following datasets: annual global net-migration rates at grid scale as a multiband GeoTIFF with 5 arc-min resolution for 2000-2019;

annual global net-migration rates for adm0, adm1 and adm2 levels as polygon layers (gpkg-files); and annual global birth and death rates as multiband GeoTIFFs with 5 arc-min resolution for 2000-2019.

Data is visualised in online net-migration explorer at https://wdrg.aalto.fi/global-net-migration-explorer/. Data underlying the web application is available in the repository with all other data.

## Human research participants

Policy information about studies involving human research participants and Sex and Gender in Research.

| | |
|---|---|
| Reporting on sex and gender | N/A |
| Population characteristics | N/A |
| Recruitment | N/A |
| Ethics oversight | N/A |

Note that full information on the approval of the study protocol must also be provided in the manuscript.

## Field-specific reporting

Please select the one below that is the best fit for your research. If you are not sure, read the appropriate sections before making your selection.

☐ Life sciences    ☒ Behavioural & social sciences    ☐ Ecological, evolutionary & environmental sciences

For a reference copy of the document with all sections, see nature.com/documents/nr-reporting-summary-flat.pdf

## Behavioural & social sciences study design

All studies must disclose on these points even when the disclosure is negative.

| | |
|---|---|
| Study description | This study utilizes quantitative demographic, environmental and socio-economic data to 1) construct a novel quantitative gridded global dataset of human net-migration, and 2) conduct a quantitative analysis describing key features and underlying structures of the produced data in relation to several demographic, socio-economic and environmental factors. |
| Research sample | All data used in this study are openly available. The existing datasets for sub-national birth and death ratios are: national censuses, EuroStat database (1), StatCompiler (2), UN databases (3) and OECD databases (4). The origin of dataset for each country is specified in the Supplementary materials (Figure S8) provided in the submission. |
| | Other datasets used include: aridity index (5), HDI (6&7), WorldPop 'population density' (8) and 'Age and sex structures' datasets (8). |
| | 1. EUROSTAT, "Live births and crude birth rate" (The statistical office of the European Union, Unit F2: Population and migration statistics, Luxembourg, 2021), (available at https://ec.europa.eu/eurostat/web/products-datasets/-/tps00204). EUROSTAT, "Deaths and crude death rate" (The statistical office of the European Union, Unit F2: Population and migration statistics, Luxembourg, 2021), (available at https://ec.europa.eu/eurostat/web/products-datasets/-/tps00029).<br>2. The DHS Program, STATcompiler (https://www.statcompiler.com/en/).<br>3. Statistics Division of the United Nations Secretariat, Standard country or area codes for statistical use (M49) - Geographic Regions. (2021), (available at https://unstats.un.org/unsd/methodology/m49/).<br>4. OECD, Mortality crude rates by cause of death, large TL2 regions, small TL3 regions (2021).<br>5. A. Trabucco, R. Zomer J., Global Aridity Index and Potential Evapo-Transpiration (ET0) Climate Database v2 (2018), (available at https://cgiarcsi.community).<br>6. M. Kummu, M. Taka, J. H. A. Guillaume, Gridded global datasets for Gross Domestic Product and Human Development Index over 1990–2015. Scientific Data. 5, 180004 (2018).<br>7. J. Smits, I. Permanyer, The Subnational Human Development Database. Scientific Data 6, 190038 (2019).<br>8. WorldPop (www.worldpop.org - School of Geography and Environmental Science, University of Southampton; Department of Geography and Geosciences, University of Louisville; Departement de Geographie, Universite de Namur), Center for International Earth Science Information Network (CIESIN), Columbia University, Global High Resolution Population Denominators Project - Funded by The Bill and Melinda Gates Foundation (OPP1134076), (available at https://dx.doi.org/10.5258/SOTON/WP00647).<br><br>Data collection and processing are described in detail in the Methods. |
| Sampling strategy | We did do sampling in the analysis as we used all available data. |
| Data collection | Data for sub-national birth and death ratios were manually collected from several sources (see above), and compiled into one spreadsheet provided as supplementary material. Missing datapoints were filled with linear interpolation and extrapolation. The procedure is described in detail in the Methods section. |

| Timing | The data collected covers years 2000-2019. |
| Data exclusions | No data were excluded in the analysis. |
| Non-participation | The study solely used existing demographic data and did not engage individuals in data collection. |
| Randomization | We used all the available data and thus no randomization was needed. |

# Reporting for specific materials, systems and methods

We require information from authors about some types of materials, experimental systems and methods used in many studies. Here, indicate whether each material, system or method listed is relevant to your study. If you are not sure if a list item applies to your research, read the appropriate section before selecting a response.

## Materials & experimental systems

| n/a | Involved in the study |
|-----|-----------------------|
| ☒ ☐ | Antibodies |
| ☒ ☐ | Eukaryotic cell lines |
| ☒ ☐ | Palaeontology and archaeology |
| ☒ ☐ | Animals and other organisms |
| ☒ ☐ | Clinical data |
| ☒ ☐ | Dual use research of concern |

## Methods

| n/a | Involved in the study |
|-----|-----------------------|
| ☒ ☐ | ChIP-seq |
| ☒ ☐ | Flow cytometry |
| ☒ ☐ | MRI-based neuroimaging |

