## [Peer Review File · Nature Human Behaviour]

Peer Review Information

Journal: Nature Human Behaviour

Manuscript Title: World's human migration patterns in 2000–2019 unveiled by high-resolution data

Corresponding author name(s): Venla Niva and Matti Kummu

Reviewer Comments & Decisions:

Decision Letter, initial version:

28th September 2022

Dear Dr. Kummu,

Thank you once again for your manuscript, entitled "World's human migration patterns in 2000-2019 unveiled by high-resolution data," and for your patience during the peer review process. I apologize for the delay in reaching a decision.

Your manuscript has now been evaluated by 3 reviewers, whose comments are included at the end of this letter. In the light of their advice, I regret that we cannot offer to publish your manuscript in Nature Human Behaviour.

While the reviewers find your work of some interest, they raise concerns about the robustness of the underlying data and the extent to which your approach sufficiently accounts for uncertainty in the underlying population estimates, as well as the strength of the conclusions that can be drawn at this stage. We feel that these reservations are sufficiently important as to preclude publication of this work in Nature Human Behaviour.

I am sorry that we cannot be more positive on this occasion but hope that you will find our reviewers' comments helpful when preparing your paper for submission elsewhere.

Sincerely,
Aisha

Aisha Bradshaw, PhD
Senior Editor
Nature Human Behaviour

Reviewer expertise:

Reviewer #1: geospatial data analysis

Reviewer #2: geography, demography, migration

Reviewer #3: geospatial data analysis

Reviewers' Comments:

Reviewer #1:

Remarks to the Author:

Key points: Please summarise what you consider to be the outstanding features of the work.

Building on an established methodology, the authors estimate net migration at up to an admin2 level on annual time steps between 2000 and 2019. The authors have made a number of improvements to the approaches developed by CIESIN and the JRC. They also show an understanding of the approaches to using globally harmonized spatial data on demographic phenomena.

Validity: Does the manuscript have flaws which should prohibit its publication? If so, please provide details.

This shouldn't prevent publication, but the authors need to be clearer about the fact that they are using modeled annual population grids that have unknown levels of certainty. This is particularly true for the 2019 data set which is probably only partially "anchored" in 2020 round censuses, since many countries have yet to hold a 2020 round census, and WorldPop almost certainly did not incorporate data for many ongoing censuses (given delays in data publication) when they compiled the 2019 data set. There are tables online showing the status of different censuses (e.g. <https://unstats.un.org/unsd/demographic-social/census/censusdates/>). The authors should provide additional information on the censuses included in the population estimates (and probably also for their birth and death rate data). And there should be some discussion of this when it comes to presentation of results for countries like Syria, where the data are highly uncertain. The bouncing around and extremes in rural and urban net migration levels in the Middle East and Eastern Europe in Figure 3 may partly be attributable to this.

Originality and significance: Are there similar resources already in existence? Is the present resource going to be useful and to whom? Is it likely to be widely adopted?

There are data sets by CIESIN and JRC that they reference, but their approach is novel and worth publishing.

Use case: Do the authors provide an empirical/practical example that demonstrates the usefulness of the resource?

Yes, and there are many other potential applications that they hint at in the discussion of results and conclusion that would be worth exploring.

Availability: Is the resource available in an easy to use format? Do the authors provide sufficient

details on how potential users can access and use the resource?

They say that the data would be made available online. The authors may wish to consider dissemination through a web site that groups other similar data such as the NASA SEDAC hosted by CIESIN. Otherwise these resources risk being fragmented over many different data repositories.

Suggested improvements: Please list additional work that could help strengthen the work in a revision.

Figure 1 is cluttered and the legends do not appear along side the maps, so this will require some improvement. The embedded text is very small. The use of the downscaling inputs c-f are not explained in the figure so their use is left to conjecture until one reads the Methods section later in the paper.

References: Does this manuscript reference previous literature appropriately? If not, what references should be included or excluded?

A few references are incomplete – e.g. reference #5

Clarity and context: Is the abstract clear, accessible? Are abstract, introduction and conclusions appropriate?

Yes.

Please indicate any particular part of the manuscript, data, or analyses that you feel is outside the scope of your expertise, or that you were unable to assess fully.

None

Details

- Fig 1: "studie period" should be "study period"
- Pg. 5: "communal level" – the different administrative levels should be referred by the more universal definition of admin0, admin1, admin2, with example terminology given for each level the first time they are introduced.
- Fig 2: It is not clear what inter-communal and inter-provincial net migration represent and it is not sufficient to refer to methods to describe what these are.
- Pg. 8: A colleague mentioned to me high rural in-migration in Cote d'Ivoire, at around the time of political instability in the early 2010s, owing to depressed economic conditions. Some greater explanation / discussion of the high positive rural net migration in certain parts of the world is warranted.
- Pg. 10: "shift turn" – one or the other

Alex de Sherbinin

Reviewer #2:

Remarks to the Author:

This is an interesting paper, however, I do have significant concerns about the potential error in the estimates. There is significant uncertainty associated with both the births and deaths data which is not

discussed in detail and will be propagated into the estimates of net migration. This will be particularly acute at the spatial scale for which estimates are produced. There also doesn't appear to be any validation of the estimates, I think there is a real danger in releasing such estimates without very strong caveats. They will be taken by many uncritical users as true.

Reviewer #3:

Remarks to the Author:

The manuscript presents the global human migration patterns at the high-resolution in both space and time. It is a meaningful work to fill the migration data gap by creating the global dataset of annual net-migration in responding to the SDGs and its findings provide insights for national policy implications. However, there are several major issues in current version.

1. I found several problems in the created annual net-migration dataset. For instance, Figure 2e, the net-migration from 2000 to 2019 is negative. But, the value may be positive during this period. Thus, authors should check the data carefully for each country in each year.

2. Again, they used the worldpop data as the reported population change. However, the accuracy of the worldpop data should be assessed before the usage for the calculation of net-migration. The necessary correction on the worldpop data may be performed. It may be one of the reasons for the inaccurate results in Figure 2.

3. I found the spatial units in Figure 2b are close to the Figure 2c. However, the spatial level between the two figures is different. It's similar for Figure 2d.

4. For the impact analysis, it is unclear for the division of the 100 socio-climatic bins. I cannot follow the idea of figure 6a and 6c.

5. They consider only one climatic factor (i.e., aridity). The reasons for this purpose are inadequate. How about other climatic factors, such as the natural disasters?

6. The reason for the study span 2000-2019 should be elaborated. I remember that Worldpop has the relevant data after 2019.

Following suitable revisions, you may want to consider transferring your manuscript. I suggest that you consider Scientific Reports as a suitable venue for your work. To transfer your manuscript there, please use our manuscript transfer portal **[LINK REDACTED]. You will not have to re-supply manuscript metadata and files, unless you wish to make modifications, but please note that this link can only be used once and remains active until used. For more information, please see our manuscript transfer FAQ page.

Note that any decision to opt in to In Review at the original journal is not sent to the receiving journal on transfer. You can opt in to In Review at receiving journals that support this service by choosing to modify your manuscript on transfer. In Review is available for primary research manuscript types only.

Author Rebuttal to Initial comments

Response to reviewers

We would like to thank the Reviewers and Editor very much for their careful reading and constructive comments. We made major revisions following the suggestions by Reviewers, including:

- 1) Validation of the data. All concerns regarding data validation were addressed. Input data, including the WorldPop population count and the births and deaths data, as well as the produced net-migration data were validated against reported statistics provided by OECD, EUROSTAT, Census data and the UN. Validation results show that the modelled and downscaled estimates are in line with the reported data.
- 2) Clarifications regarding the administrative units. We have now addressed all concerns and added clarifications regarding the definitions of each unit.
- 3) Clarifications regarding the analysis. We have now added clarification regarding the division of bins in the socio-climatic analysis as well as elaborated the selection of variables.

Further, we put together an online tool that allows a user to explore our net-migration, births, deaths and population change data at year-by-year basis across the three administrative levels used for the analyses (admin 0 level, i.e. country; admin 1 level, i.e. provincial; and admin 2 level, i.e. communal): <http://193.166.24.46:3838/migrationShiny/R/>

These and other revisions are detailed below in our point-by-point replies to the comments. All the changes in the manuscript are highlighted in the tracked version of the manuscript. The updated line numbers (from the clean revised document) are reported in the responses where applicable.

Reviewer #1:

Key points: Please summarise what you consider to be the outstanding features of the work.

Comment 1.1. Building on an established methodology, the authors estimate net migration at up to an admin2 level on annual time steps between 2000 and 2019. The authors have made a number of improvements to the approaches developed by CIESIN and the JRC. They also show an understanding of the approaches to using globally harmonized spatial data on demographic phenomena.

Reply 1.1. Thank you for the nice summary about the novelty of our work.

Validity: Does the manuscript have flaws which should prohibit its publication? If so, please provide details.

C1.2. This shouldn't prevent publication, but the authors need to be clearer about the fact that they are using modeled annual population grids that have unknown levels of certainty. This is particularly true for the 2019 data set which is probably only partially "anchored" in 2020 round censuses, since many countries have yet to hold a 2020 round census, and WorldPop almost certainly did not incorporate data for many ongoing censuses (given delays in data publication) when they compiled the 2019 data set. There are tables online showing the status of different censuses (e.g. <https://unstats.un.org/unsd/demographic-social/census/censusdates/>).

The authors should provide additional information on the censuses included in the population estimates (and probably also for their birth and death rate data). And there should be some discussion of this when it comes to presentation of results for countries like Syria, where the data are highly uncertain. The bouncing around and extremes in rural and urban net migration levels in the Middle East and Eastern Europe in Figure 3 may partly be attributable to this.

R1.2: Very important point. We agree that the validations of the data were missing, and have now included the following validation results in the manuscript and the supplementary material:

Modelled population grids (i.e. WorldPop grids) were validated against observed population counts in admin 2 level OECD units (communal level) (n= 1818). Validation results for years 2000, 2005, 2010, 2015 and 2020 show strong and significant ($p < 0.001$) correlations between observed and modelled values (Pearson's $R=0.99$). These WorldPop validation results are now presented in the Supplements in Figure S11. We also cross-validated the WorldPop data with another global gridded population datasets, LandScan (<https://landscan.ornl.gov/>), for year 2010. We extracted the population count to the admin 1 level (n=2460) and compared these data. We found that the agreement was very strong (Pearson's $R = 1$, $p < 2.2-16$) (see Figure in R3.3). We also conducted validation analyses for births, deaths and net-migration. See our reply to Comment 1 of the reviewer 2.

Additional information (such as the description, resolution, timespan and the source) of the census data collected, for both births and deaths, are provided in the Supplementary data file provided in the submission along with the raw subnational birth and death data for all years. We have now clarified this also in the manuscript (L103): "Additional information of the collected census data, such as the description, resolution, timespan and sources, are provided in the Supplementary Data file."

Further, the case of Syria is discussed on L206: "In Eastern Europe and the Middle East, for instance, rural and urban net-migration rates fluctuated annually, especially during the years preceding the Arab spring, followed by massive rural out-migration and urban in-migration between 2011–2013 (Figure 3e). Out-migration from Syria was among the largest in the world between 2010–15, when more than 2 million people left to neighbouring Turkey and Lebanon(2, 15, 25). This explains a sharp influx of migrants in rural and urban areas in Eastern Europe group (including Turkey) (Figure 3c)." We also now added a disclaimer reminding the reader about the uncertainties of the data, especially in these regions (L212): "Despite our results align with

previous estimates of migration in the Middle East, it should be noted that these regions are prone to high uncertainties in regards to data.” We thank the reviewer for bringing this up.

C1.3. Originality and significance: Are there similar resources already in existence? Is the present resource going to be useful and to whom? Is it likely to be widely adopted?

There are data sets by CIESIN and JRC that they reference, but their approach is novel and worth publishing.

R1.3: Thank you for the recognition of the novelty and contribution of our work.

C1.4. Use case: Do the authors provide an empirical/practical example that demonstrates the usefulness of the resource?

Yes, and there are many other potential applications that they hint at in the discussion of results and conclusion that would be worth exploring.

R1.4: Thank you for the comment.

C1.5. Availability: Is the resource available in an easy to use format? Do the authors provide sufficient details on how potential users can access and use the resource?

They say that the data would be made available online. The authors may wish to consider dissemination through a web site that groups other similar data such as the NASA SEDAC hosted by CIESIN. Otherwise these resources risk being fragmented over many different data repositories.

R1.5: We agree and are happy to disseminate our data through the appropriate channels.

C1.6. Suggested improvements: Please list additional work that could help strengthen the work in a revision.

Figure 1 is cluttered and the legends do not appear along side the maps, so this will require some improvement. The embedded text is very small. The use of the downscaling inputs c-f are not explained in the figure so their use is left to conjecture until one reads the Methods section later in the paper.

R1.6: The Figure is now revised as suggested.

C1.7. References: Does this manuscript reference previous literature appropriately? If not, what references should be included or excluded?

A few references are incomplete – e.g. reference #5

R1.7: Thanks for pointing this out. References (#5, #31, #49, #61) are now fixed.

C1.8. Clarity and context: Is the abstract clear, accessible? Are abstract, introduction and conclusions appropriate?

Yes.

R1.8: Thank you for the comment.

C1.9. Please indicate any particular part of the manuscript, data, or analyses that you feel is outside the scope of your expertise, or that you were unable to assess fully.

None

R1.9: Thank you for the comment.

Details

C1.10. - Fig 1: “studie period” should be “study period”: R1.10: Fixed.

C1.11. - Pg. 5: “communal level” – the different administrative levels should be referred by the more universal definition of admin0, admin1, admin2, with example terminology given for each level the first time they are introduced.

R1.11: Agreed. Now revised as follows on L186: *“We assessed if this held true globally across 12 regions and for each country at three administrative levels by using the GADM delineation (national = admin0, provincial = admin1 and communal = admin2).”*

This was fixed also for Introduction, L71: *“It is also possible to analyse the type of sending and receiving areas (i.e. rural or urban) at multiple scales (regional scale and three administrative levels (GADM delineation): national (admin0), provincial (admin1) and communal (admin2)) over the past two decades.”*

C1.12. - Fig 2: It is not clear what inter-communal and inter-provincial net migration represent and it is not sufficient to refer to methods to describe what these are.

R1.12: Agreed. We have now revised the analysis presented in Figure 2. It now describes the trends of net-migration over the study period on the three administrative scales. The analysis and results are shown on L160: *“We assessed this by calculating how net-migration at each level has changed over the study period by using linear regression. The results follow a similar pattern as cumulative net-migration where the trend changes according to the level...”*

The caption of Fig 2 was also revised accordingly.

C1.13. - Pg. 8: A colleague mentioned to me high rural in-migration in Cote d’Ivoire, at around the time of political instability in the early 2010s, owing to depressed economic conditions. Some greater explanation / discussion of the high positive rural net migration in certain parts of the world is warranted.

R1.13: Agreed. We now discuss this on L235: *“Extensive rural in-migration is most likely explained by the inter-provincial and inter-communal migrations between rural areas and immigration from other countries. Studies show that a trend of rural-urban migration is shifting towards more complex mobility patterns, of which rural-rural mobility is one of the most prevalent types of internal migration. Especially in Sub-Saharan Africa people tend to move between rural areas as seasonal circular migration and economic diversification given after better access to land or better job prospects than in cities (26, 27). In Europe, similar pattern appears in rural areas which attract workers in the agricultural sector within the same country or from abroad (28). Large urban agglomerates may also push people to move to rural areas in search for more affordable housing (counter-urbanization) (See Buckle and Osbaldiston (29) for counter-urbanization in Australia and Clark (30) for the US).”*

(26): Mercandalli S., Losch B. (eds.), Belebema M.N., Bélières J.-F., Bourgeois R., Dinbabo M.F., Fréguin-Gresh S., Mensah C., Nshimbi C.C. 2019. Rural migration in sub-Saharan Africa: Patterns, drivers and relation to structural transformation. Rome, FAO and CIRAD. <https://doi.org/10.4060/ca7404en>

(27): M. Steinbrink, H. Niedenführ, *Africa on the Move: Migration, Translocal Livelihoods and Rural Development in Sub-Saharan Africa* (Springer International Publishing, 2020) <https://doi.org/10.1007/978-3-030-22841-5> (February 8, 2023).

(28): F. Natale, S. Kalantaryan, M. Scipioni, A. Alessandrini, A. Pasa, *Migration in EU rural areas* (Publications Office of the European Union, 2019) (January 25, 2023).

(29): Caitlin Buckle & Nick Osbaldiston (2022) Editorial introduction: counter-urbanisation in contemporary Australia: a review of current issues and events, *Australian Geographer*, 53:4, 347-362, DOI: 10.1080/00049182.2022.2137902

(30): S. L. Clark, *In Search of Housing: Urban Families in Rural Contexts**. *Rural Sociology* 77, 110–134 (2012).

C1.14. - Pg. 10: “shift turn” – one or the other

R1.14: Fixed.

Alex de Sherbinin

Reviewer #2:

Remarks to the Author:

Comment 2.1. This is an interesting paper, however, I do have significant concerns about the potential error in the estimates. There is significant uncertainty associated with both the births and deaths data which is not discussed in detail and will be propagated into the estimates of net migration. This will be particularly acute at the spatial scale for which estimates are produced. There also doesn't appear to be any validation of the estimates, I think there is a real danger in releasing such estimates without very strong caveats. They will be taken by many uncritical users as true.

R2.1: Thank you for pointing this out, and we agree that we did communicate the uncertainties of the data inadequately in the original submission. In the revised manuscript we now provide validation analyses as well as aim to communicate the uncertainty of the data better.

First, we collected substantial sub-national (admin 1 level) database of both death and birth data. This is the first database of such an extensive sub-national collection of census and other data. We do acknowledge that these sub-national datasets are from various sources and sometimes reported slightly differently. And that downscaling would further introduce uncertainty. To minimise the uncertainty, the produced birth and death grids were adjusted so

that the subnational sum of births and deaths matched with the collected subnational (admin 1 level) birth and death counts (Supplementary data). And these subnational data at admin 1 level were harmonised with the national data; i.e. we ensured that the sum of births and deaths at the grid level agree at the national level with UN statistics.

We now explicitly mention this in the text too; please see L430: *“we adjusted the downscaled approximations to ensure that the total number of births/deaths represented in the downscaled dataset matched the collated census data at the subnational scale”*. We also made this more prominent in the Results section (L95): *“This here-developed dataset (openly available at: [will be made publicly available upon publication]) was constructed from subnational (admin 1 level) birth and death rate data collected across 2,567 and 2,276 administrative units, respectively (Figure 1a–b; Methods), and downscaled to 5 arc-min resolution with rasterised socio-economic data developed in this study, and finally adjusted to match the subnational data collected (Figure 1c–f; see Methods for details).”*

We further validated the downscaled birth and death data using admin 2 level subnational data, i.e. more detail level data than used to create the downscaled dataset for which we used admin 1 level data. Downscaled births and deaths data were validated with this admin 2 level data from EUROSTAT data (n=1504) and OECD (n=1883), respectively. Validation results show strong and significant ($p < 0.001$) correlation between the observed and downscaled values (births: $R=0.81$; deaths: $R=0.78$). We note that the validation data from OECD and EUROSTAT databases are not representative of data from data scarce regions, like Africa or Middle East, where census reporting entails more uncertainties than the above sources. However, harmonizing the gridded data to match the subnational (admin 1 level) data used for producing the grids prevents spreading the downscaled values out from admin 1 level areas, i.e. census values always hold at the admin 1 level. Residual uncertainty at this level is then due to census reporting and not our downscaling method. In addition – where data availability allows – the validation results at a more detailed (admin 2 level) scale provide ample evidence that our downscaling method is reasonable. Therefore, we argue that our data processing and validation approaches are sufficiently rigorous and faithful to preserving reported census data. Validation results for births and deaths are provided in the Supplements in Figure S12.

Validation of the WorldPop (see our reply to comment 1.2 of the Reviewer 1 and reply to comment 3.3 of the Reviewer 3), births and deaths data indicate that the net-migration data represents observed net-migration patterns well. We also validated our net-migration data with national net-migration stocks provided by the UN and then with subnational data from the US (admin 2 level), Europe (admin 1 level) and South Korea (admin 1 level). This was done by aggregating the developed net-migration data for each admin unit and for each year and for 5 and 10 year cumulative values. Then our net-migration data was compared to the reported values. National and subnational level comparisons are shown in Figures S13-16 and in Table S2. The results show that at national level there is a strong and statistically significant correlation between our estimated values and observed values (for example for year 2010, Pearson's $R = 0.72$). Subnational results show more variation but indicate an overall good and very good correlations between the estimated and reported net-migration ratio (NM per 1000 people) (for example for 2001-2010 – for which reported data for all areas were available – the correlations

are as follows: Pearson’s $R = 0.95$ for USA, $R = 0.46$ for EU and $R = 0.97$ for South Korea). Some of the differences in the estimated and reported sub-national net-migration counts (in Europe for instance) can be explained by different definition and data used for estimating net-migration counts. We elaborate on this in the Supplements.

Validation methods are described in L713 in the manuscript and in the Supplements with more details.

Further, we put together an online explorer, where it is possible to explore the annual migration, births, deaths and population change for any given unit at all three admin levels (country, province, communal). It can be accessed through the link below:

<http://193.166.24.46:3838/migrationShiny/R/>

Finally, we added short disclaimer to the article that the people should be aware of the uncertainty when using the data (L104): “We note that some of the collected and produced data are prone to uncertainties which should be taken into account when using the data.”

Figure 1. Left: Validation of (a) births and (b) death data at subnational level 3 using EUROSTAT and OECD data, respectively.

Reviewer #3:

Remarks to the Author:

Comment 3.1 The manuscript presents the global human migration patterns at the high-resolution in both space and time. It is a meaningful work to fill the migration data gap by creating the global dataset of annual net-migration in responding to the SDGs and its findings provide insights for national policy implications. However, there are several major issues in current version.

Reply 3.1: Thank you for highlighting the contribution of our work in bridging the data gap. We have addressed all the issues the reviewer raised below.

C3.2: I found several problems in the created annual net-migration dataset. For instance, Figure 2e, the net-migration from 2000 to 2019 is negative. But, the value may be positive during this period. Thus, authors should check the data carefully for each country in each year.

R3.2: Thanks for pointing this out. First, we have double checked all the data and everything should be in order. Net-migration grids were validated at national level (n=238) by aggregating gridded net-migration for each country, calculating the ratio (per 1000 people) and then comparing the count with net-migration ratio provided by the UN. The validation results show a strong and significant ($p < 0.001$) correlation between the estimated and UN reported net-migration ratios (Pearson's R = 0.75, 0.9, 0.61, 0.69 for years 2001, 2005, 2010 and 2015 respectively. Validation results are provided in the Supplements in FigS13 and Table S2.

We also validated the net-migration results with the reported subnational data from the US (admin 2 level), Europe (admin 1 level) and South Korea (admin 1 level). This was done by aggregating the gridded net-migration data for each admin unit, for each year and for 5- and 10-year cumulative values. Then our net-migration data was compared to the reported values provided by the UN and other sources. National and subnational level comparisons are shown in Figures S13, S13, S15, S16 and in Table S2. The results show that at national level there is a strong and statistically significant correlation between the estimated and observed values (for example for year 2010, Pearson's R = 0.72). Subnational results show more variation but indicate an overall good and very good correlations between the estimated and reported net-migration counts and ratios (for example for 2000-2010 – for which reported data for all areas were available – the correlations are as follows: Pearson's R = 0.95 for USA, R = 0.46 for EU and R = 0.97 for South Korea). Some of the differences in the estimated and reported sub-national net-migration counts (in Europe for instance) can be explained by different definition and data used for estimating net-migration counts. We elaborate on this in the Supplements.

Validation methods are described in L713 in the manuscript and in the Supplements with more details.

Yes, reviewer is correct that the Fig 2e shows the cumulative net-migration over the 20 years and there might be years with positive net-migration in a country where cumulative net-migration is negative. Due to limit of space, we are not able to report the net-migration for each year. However, to make exploring our results easier, we put together an online tool, in which it is possible to visualise annual net-migration, deaths, births and population change for any given unit at all three scales (admin 0 level, admin 1 level, admin 2 level). It can be accessed through the link below:

<http://193.166.24.46:3838/migrationShiny/R/>

We hope that this tool will facilitate the Reviewer's work in assessing the quality of the data.

C3.3: Again, they used the worldpop data as the reported population change. However, the accuracy of the worldpop data should be assessed before the usage for the calculation of net-migration. The necessary correction on the worldpop data may be performed. It may be one of the reasons for the inaccurate results in Figure 2.

R3.3: Good point. Prior to any analysis the WorldPop data was harmonised so that the national level population counts matched the UN reported national level population counts.

We also validated the WorldPop population grids against observed population counts in admin 2 level (communal level) OECD units (n= 1818). Validation results for 2000, 2005, 2010, 2015 and 2020 show strong and significant ($p < 0.001$) correlations between observed and modelled values (Pearson's $R=0.99-1$, depending on the year). These WorldPop validation results are now presented in the Supplements in Figure S11.

Finally, we also cross-validated the WorldPop data with another global gridded population datasets, LandScan (<https://landscan.ornl.gov/>), for year 2010. We extracted the population count to the admin 1 level (n=2460) and compared these data. We found that the agreement was very strong (Pearson's $R = 1$, $p < 2.2 \times 10^{-16}$).

Thus, we are confident that the underlying population dataset is suitable for our net-migration estimates.

C3.4: I found the spatial units in Figure 2b are close to the Figure 2c. However, the spatial level between the two figures is different. It's similar for Figure 2d.

R3.4: Thanks, we agree this should have been clearer. We have now revised the Figure 2 and the analysis. Now Figures 2b, d and f illustrate the trend of net-migration ratio (net-migration per 1000 people) over 2000-2019, and the administrative levels are aligned with Figures 2a, c and e which illustrate the cumulative net-migration per 1000 people.

C3.5: For the impact analysis, it is unclear for the division of the 100 socio-climatic bins. I cannot follow the idea of figure 6a and 6c.

R3.5: Thanks for pointing this out. We agree that this needs further clarification. The Figure shows urban and rural net-migration (per urban/rural pop) in each bin as heatmaps and

geographic maps. Here, the geographic maps are spatial representations of the heatmaps, i.e. each cell in the heatmap corresponds to a socio-climatic zone (a polygon) in the geographic map.

Bins are determined based on socio-economic and climatic conditions and weighted by total population in each bin. Due to this, each bin accommodates 1% of the global population. Bins were determined by first dividing all grid cells of the HDI data into 10 quantiles. Then the grid cells in the aridity raster within each HDI quantile were divided into 10 quantiles. Thus, in the first bin (bin no. 0) would be all values located in the first aridity quantile of the first HDI quantile, while in the second bin (bin no. 1) would be all values located in the second aridity quantile of the first HDI quantile, etc. We have now added a schematic illustration of how the bins are divided in the Supplements (Fig S3). By using our urban-rural classification, we could then calculate urban and rural net-migration in each bin.

We have now added clarification on the division of the bins in the caption of Figure 6, L333: *“Figure 6. (a) Urban net-migration per 1000 urban population and (b) the impact of urban net-migration on urban population change in each socio-climate bin, (c) Rural net-migration per 1000 rural population and (d) the impact of rural net-migration on rural population change in each climate bin. Here, net-migration per urban/rural population was calculated as a zonal sum over net-migration in urban (a) and rural (c) areas in each bin, and then divided with the respective urban/rural population count in the respective bin. The maps are spatial representations of the heatmaps. Socio-climatic bins are based on socio-economic and climatic conditions (see Methods) and defined by using Human Development and aridity indices.”*

As well as in the Methods, L808: *“Our binning divides global inhabited areas into 100 socio-climatologically analogous zones, that have similar human development and climatic conditions. The binning was conducted in two steps. First, we divided all considered grid cells into 10 population-weighted quantiles based on HDI. After that, each HDI quantile was again divided into 10 population-weighted quantiles based on aridity. All values located in the first aridity quantile of the first HDI quantile would fall under the first bin (bin no. 0), while the values located in the second aridity quantile of the first HDI quantile would go in the second bin (bin no. 1). The division of bins is illustrated in Figure S3. This division ensured that each bin incorporates 1% of the global population. The binning represented as a heatmap could be then transformed into a map representation (See FigS2c). Urban and rural areas in each bin could be then extracted by using the urban extent data (see above). Global maps and description of the pre-processing of both human development, aridity index and socio-climatic bins are available in the Supplements (Figure S2). “*

C3.6: They consider only one climatic factor (i.e., aridity). The reasons for this purpose are inadequate. How about other climatic factors, such as the natural disasters?

R3.6: Thanks for the valid question. We agree that natural disasters would be an interesting variable if we had assessed interannual variability of net-migration, for instance. However, here we used aridity because it combines temperature and precipitation, and is hence a good proxy for climatological conditions, particularly average long-term dryness which is relevant for the ecosystems and economic sectors. In this analysis, the net-migration was treated as a long-term sum, or in other words, accumulated net-migration over the study period. Thus, using natural

disasters, which represent a short-term disruption in the environment, as a variable with 20-years accumulated net-migration would not be feasible.

We have now elaborated this in the Methods, L800: *“For the climatic factor, aridity index – a ratio of potential evaporation to precipitation – was chosen to capture different atmosphere and land surface processes shaping terrestrial dryness. The aridity index has been used to assess desertification under climate change (74). Here the zonal analyses were conducted by using a long-term estimate of net-migration (accumulated sum over the study period, 2000-2019), which is why the impact of a short-term variable, such as natural disasters, would not be visible in the analysis.”*

We further elaborate the limitations of using net-migration as an estimate of human mobility in the context of environmental change and extremes in the discussion L416: *“First, given that our data derive net-migration from the difference between total and natural population change, it is impossible to distinguish between different types of migrants such as refugees, internally displaced persons or economic migrants. Our data mask out important individual migration events and external shocks such as conflict altogether by only describing whether an area experienced more in-migration than out-migration, or vice versa, without being able to differentiate migration flows in and/or out of an area (47). Further, our analysis utilised long-term averages of climatic and socio-economic conditions, thus masking any interannual or intra-annual variation of net-migration related to sudden shocks such as extreme weather events.”*

C3.7: The reason for the study span 2000-2019 should be elaborated. I remember that Worldpop has the relevant data after 2019.

R3.7: We agree that the selection of the study period should be justified better. The study span from 2000-2019 was selected based data availability. In constructing the data, we have used the WorldPop data up to year 2020, the latest available data as of the date of data processing. In order to compute net-migration for a given year, we calculated the population change between the year in question as well as the following year, in addition to the natural change calculated from birth and death data. For instance, the population count from year 2020 was needed to calculate the population change between 2020 and 2019 in order to determine net-migration for year 2019. Currently, the WorldPop has data for years 2000-2020, thus years after that could not be incorporated in the dataset.

Decision Letter, first revision:

10th May 2023

Dear Dr. Kummu,

Thank you for submitting your revised manuscript "World's human migration patterns in 2000-2019 unveiled by high-resolution data" (NATHUMBEHAV-22071674A-Z). It has now been seen by the original referees and their comments are below. As you can see, the reviewers find that the paper has

improved in revision. We will therefore be happy in principle to publish it in Nature Human Behaviour, pending minor revisions to satisfy the referees' final requests and to comply with our editorial and formatting guidelines.

We are now performing detailed checks on your paper and will send you a checklist detailing our editorial and formatting requirements within ten days. Please do not upload the final materials and make any revisions until you receive this additional information from us.

Sincerely,
Aisha

Aisha Bradshaw, PhD
Senior Editor
Nature Human Behaviour

Reviewer #1 (Remarks to the Author):

With one caveat, the authors appear to have addressed all the review comments appropriately. The paper makes a significant contribution to migration data availability. The caveat: Both reviewers hone in on uncertainty. Validating WorldPop against OECD data and obtaining a very high correlation is not really addressing the issue, since there is much higher uncertainty in all the non-OECD countries. I think the authors simply need to be more forthright in stating that census data availability in many countries (e.g. much of Africa) is very poor and that uncertainties in those country data will propagate through in the NM estimates. The current statement "We note that some of the collected and produced data are prone to uncertainties which should be taken into account when using the data" needs to be qualified more, to address *where* the uncertainties are highest.

There are a few minor points that require attention (below), and the ms should be thoroughly proofread for typos and English language usage issues.

Figure 1: "cencuses" should be "censuses"; "was done using multiple" should be "was done using a multiple"; "here shown" should be "the depicted"

The figure and elsewhere in the ms the authors speak of birth/death ratio rather than birth/death rate. I think "rate" is the more appropriate term, since a ratio is a comparison of two numbers or measurements (i.e., the sex ratio is the number of males divided by the number of females).

Figure 6: In the caption: There are two "(b)" labels. "sum over" should be "some of" no? Figure: I was confused by the "Aridity" label on the charts since dry is to the left and wet is to the right in the key, which means aridity declines to the right. I suggest another label such as "Wetness" or reversing the the directionality.

Reviewer #2 (Remarks to the Author):

While I still have some concerns about the uncertainty in these estimates, I believe that the reviewers have addressed some of my concerns about both the uncertainty and validation of estimates. The inclusion of the online tool is very useful. I am happy to recommend acceptance of the manuscript.

Reviewer #3 (Remarks to the Author):

Authors addressed most of my concerns and the revised version improves deeply. There are also several issues needed to further clarify.

1. For the comment C3.3, they made a good validation to demonstrate the accuracy. However, the Person's R between worldpop and landscan is 1, it's impossible according to the scatter plot as they have difference. $R=1$ needs two variables completely equal. They may use the rounding numbers for representing R, but it should revise. It's also for Figure S11.

2. For the comment C3.6, I agree that natural disasters have a short impact on migration. It's interesting for me that how wars impact on the migration. I hope authors add insights for the migration in several countries where had happened wars, such as Afghanistan, Iraq, Syria, etc.

3. For the comment C3.7, COVID-19 may be one of the reasons and authors may add it.

4. I recommend authors add a citation for the first presence of the area-to-point kriging (ATPK) in lines 662-663.

Author Rebuttal, first revision:

Response letter to reviewers' comments

Reviewer 1

Comment 1.1: With one caveat, the authors appear to have addressed all the review comments appropriately. The paper makes a significant contribution to migration data availability. The caveat: Both reviewers hone in on uncertainty. Validating WorldPop against OECD data and obtaining a very high correlation is not really addressing the issue, since there is much higher uncertainty in all the non-OECD countries. I think the authors simply need to be more forthright in stating that census data availability in many countries (e.g. much of Africa) is very poor and that uncertainties in those country data will propagate through in the NM estimates. The current statement "We note that some of the collected and produced data are prone to uncertainties which should be taken into account when using the data" needs to be qualified more, to address *where* the uncertainties are highest.

Reply 1.1: First, thank you very much for your positive comments on the revised manuscript. We do agree that it is very important to clearly communicate the uncertainty in the data and the sources of it. When revising the manuscript, we added one paragraph where we aim to clearly address this issue. See lines 105-

"It should be noted that our data are prone to uncertainties that originate from collected sub-national data but propagate to all derived data products – including birth and death rates, natural change in population, as well as net-migration estimates. Subject to data availability, we performed a partial validation for our data products by comparing gridded data with sub-national (mostly admin 2 level) reported data (Methods, Supplementary materials, Table S2). However, this validation cannot capture areas in which uncertainties may be the highest – i.e. areas in which the collected census data is of poor availability or quality, suffering from e.g. sporadic census years or changing sub-national administrative units. To ensure global spatial and temporal coverage, we applied a series of adjustments and corrections to the data (Methods). Nevertheless, higher uncertainties remain in some countries (such as those in Africa, the Middle East, and parts of Asia) than in others (such as those in Europe and much the Americas). As proxies of original data quality, we provide the description, resolution, timespan, and sources of each collected data set in the Supplementary Data file."

C1.2: There are a few minor points that require attention (below), and the ms should be thoroughly proofread for typos and English language usage issues.

R1.2: We have addressed the minor points, as detailed below in our responses. Further, the article has been proofread.

C1.3: Figure 1: "cencuses" should be "censuses"; "was done using multiple" should be "was done using a multiple"; "here shown" should be "the depicted"

R1.3: Thank you! Figure was modified accordingly.

C1.4: The figure and elsewhere in the ms the authors speak of birth/death ratio rather than birth/death rate. I think "rate" is the more appropriate term, since a ratio is a comparison of two numbers or measurements (i.e., the sex ratio is the number of males divided by the number of females).

R 1.4: Thank you, this is a very good comment. We have now replaced "ratio" with

“rate” in all appropriate occasions.

C1.5: Figure 6: In the caption: There are two "(b)" labels. "sum over" should be "some of" no? Figure: I was confused by the "Aridity" label on the charts since dry is to the left and wet is to the right in the key, which means aridity declines to the right. I suggest another label such as "Wetness" or reversing the the directionality.

R1.5: Caption: we have now revised the caption accordingly. Figure: we agree and changed the “Aridity” to “Humidity”.

Reviewer 2

Comment 2.1: While I still have some concerns about the uncertainty in these estimates, I believe that the reviewers have addressed some of my concerns about both the uncertainty and validation of estimates. The inclusion of the online tool is very useful. I am happy to recommend acceptance of the manuscript.

Reply 2.1: We understand well the reviewer’s concern about the uncertainty. We have now aimed to address the uncertainty even more clearly in the revised manuscript and added one paragraph of it to the main text (see lines 105-).

“It should be noted that our data are prone to uncertainties that originate from collected sub-national data but propagate to all derived data products – including birth and death rates, natural change in population, as well as net-migration estimates. Subject to data availability, we performed a partial validation for our data products by comparing gridded data with sub-national (mostly admin 2 level) reported data (Methods, Supplementary materials, Table S2). However, this validation cannot capture areas in which uncertainties may be the highest – i.e. areas in which the collected census data is of poor availability or quality, suffering from e.g. sporadic census years or changing sub-national administrative units. To ensure global spatial and temporal coverage, we applied a series of adjustments and corrections to the data (Methods). Nevertheless, higher uncertainties remain in some countries (such as those in Africa, the Middle East, and parts of Asia) than in others (such as those in Europe and much the Americas). As proxies of original data quality, we provide the description, resolution, timespan, and sources of each collected data set in the Supplementary Data file.”

Reviewer 3

Comment 3.0: Authors addressed most of my concerns and the revised version improves deeply. There are also several issues needed to further clarify.

Reply 3.0: thank you very much for your kind comment. We have further clarified the issues commented on by the reviewer, as detailed below in our point-by-point responses.

C3.1. For the comment C3.3, they made a good validation to demonstrate the accuracy. However, the Person's R between worldpop and landscan is 1, it's impossible according to the scaWer plot as they have difference. $R=1$ needs two variables completely equal. They may use the rounding numbers for representing R, but it should revise. It's also for Figure S11.

R3.1: We agree. We have now revised Supplementary Figure 1 (old Figure S11) and updated the rounded Person's R to elsewhere too.

C3.2. For the comment C3.6, I agree that natural disasters have a short impact on migration. It's interesting for me that how wars impact on the migration. I hope authors add insights for the migration in several countries where had happened wars, such as Afghanistan, Iraq, Syria, etc.

R3.2: We have briefly addressed the relationship between conflict and migration in

lines 128: *"Net-migration was negative in countries like Syria, Lithuania, Zimbabwe, Venezuela and Guyana (Figure 2e) – in line with previous assessments from Venezuela and Syria, where millions of people have fled a humanitarian crisis and conflict(15–17)"*

and lines 178: *" In Eastern Europe and the Middle East, rural and urban net-migration rates fluctuated annually, especially during the years preceding the Arab spring, followed by massive rural out-migration and urban in-migration between 2011–2013 (Figure 3e). Out-migration from Syria was among the largest in the world between 2010–15, when more than 2 million people let to neighbouring countries Turkey and Lebanon(2, 15, 24)."*

Given high uncertainties related to producing net-migration data based on censuses in conflict regions, as well as the inability of net-migration to differentiate in- and out-migration events, we are hesitant to explicitly claim conflicts driving migration, besides the above-mentioned country scale examples that are supported by other literature. However, we acknowledge that conflicts are a strong driver of migration (although our net-migration data may mask how strong) and now write in lines 340- (addition bolded):

“Our data mask out important individual migration events and external shocks such as conflict altogether by only describing whether an area experienced more in-migration than out-migration, or vice versa, without being able to differentiate migration flows in and/or out of an area (46). This may be highlighted, for instance, in areas under prolonged conflicts during our study period, such as Afghanistan, Syria, and Iraq.”

C3.3. For the comment C3.7, COVID-19 may be one of the reasons and authors may add it.

R3.3: The comment C3.7 of the previous review round considered elaborating the study span 2000–2019, and we are therefore unsure of the intention of this comment. We elaborated on the study span in the response on 1st revision round as follows: *“The study span from 2000-2019 was selected based data availability. In constructing the data, we have used the WorldPop data up to year 2020, the latest available data as of the date of data processing. In order to compute net-migration for a given year, we calculated the population change between the year in question as well as the following year, in addition to the natural change calculated from birth and death data. For instance, the population count from year 2020 was needed to calculate the population change between 2020 and 2019 in order to determine net-migration for year 2019. Currently, the WorldPop has data for years 2000-2020, thus years after that could not be incorporated in the dataset.”*

However, we seem to have forgoWen to add an explicit text addition detailing this, for which we apologise. We have now added to lines 394-:

“Although WorldPop data were available also for the year 2020 at the time of data processing, our analysis required computing population change that could not be done for the 2020 WorldPop data, and thus, final data are limited to 2019.”

C3.4. I recommend authors add a citation for the first presence of the area-to-point kriging (ATPK) in lines 662-663.

R3.4: We added the following references there:

D. Murakami and M. Tsutsumi, "Area-to-point parameter estimation with geographically weighted regression", *J. Geograph. Syst.*, vol. 17, pp. 207-225, Jul. 2015.

P. C. Kyriakidis, "A geostatistical framework for area-to-point spatial interpolation", *Geograph. Anal.*, vol. 36, no. 3, pp. 259-289, 2004.

Final Decision Letter:

Dear Professor Kummu,

We are pleased to inform you that your Resource "World's human migration patterns in 2000-2019 unveiled by high-resolution data", has now been accepted for publication in *Nature Human Behaviour*.

Please note that *Nature Human Behaviour* is a Transformative Journal (TJ). Authors whose manuscript was submitted on or after January 1st, 2021, may publish their research with us through the traditional subscription access route or make their paper immediately open access through payment of an article-processing charge (APC). Authors will not be required to make a final decision about access to their article until it has been accepted. IMPORTANT NOTE: Articles submitted before January 1st, 2021, are not eligible for Open Access publication. Find out more about Transformative Journals

Acceptance of your manuscript is conditional on all authors' agreement with our publication policies (see <http://www.nature.com/nathumbehav/info/gta>). In particular your manuscript must not be published

elsewhere and there must be no announcement of the work to any media outlet until the publication date (the day on which it is uploaded onto our web site).

With best regards,
Aisha

Aisha Bradshaw, PhD
Senior Editor
Nature Human Behaviour